# Instance-adaptive Zero-shot Chain-of-Thought Prompting

**Xiaosong Yuan**[1,2†], **Chen Shen**[3‡], **Shaotian Yan**[3], **Xiaofeng Zhang**[4]
**Liang Xie**[5], **Wenxiao Wang**[6], **Renchu Guan**[1,2], **Ying Wang**[1,2*], **Jieping Ye**[3]
[1]College of Computer Science and Technology, Jilin University
[2]Key Laboratory of Symbolic Computation and Knowledge Engineering, MoE, Jilin University
[3]Alibaba Cloud Computing
[4]Shanghai Jiao Tong University
[5] College of Computer Science and Technology, Zhejiang University of Technology
[6]College of Software, Zhejiang University
yuanxs19@mails.jlu.edu.cn, zjushenchen@gmail.com

## Abstract

Zero-shot Chain-of-Thought (CoT) prompting emerges as a simple and effective strategy for enhancing the performance of large language models (LLMs) in real-world reasoning tasks. Nonetheless, the efficacy of a singular, task-level prompt uniformly applied across the whole of instances is inherently limited since one prompt cannot be a good partner for all, a more appropriate approach should consider the interaction between the prompt and each instance meticulously. This work introduces an instance-adaptive prompting algorithm as an alternative zero-shot CoT reasoning scheme by adaptively differentiating good and bad prompts. Concretely, we first employ analysis on LLMs through the lens of information flow to detect the mechanism under zero-shot CoT reasoning, in which we discover that information flows from question to prompt and question to rationale jointly influence the reasoning results most. We notice that a better zero-shot CoT reasoning needs the prompt to obtain semantic information from the question, and then the rationale aggregates sufficient information from the question directly and via the prompt indirectly. On the contrary, lacking any of those would probably lead to a bad one. Stem from that, we further propose an instance-adaptive prompting strategy (IAP) for zero-shot CoT reasoning. Experiments conducted with LLaMA-2, LLaMA-3, and Qwen on math, logic, and commonsense reasoning tasks (e.g., GSM8k, MMLU, Causal Judgement) obtain consistent improvement, demonstrating that the instance-adaptive zero-shot CoT prompting performs better than other task-level methods with some curated prompts or sophisticated procedures, showing the significance of our findings in the zero-shot CoT reasoning mechanism.

## 1 Introduction

Large language models (LLMs) have demonstrated capabilities at tackling copious reasoning tasks through Chain-of-Thought (CoT) [1–10]. Compared to the few-shot setting for CoT generally, zero-shot CoT prompting can achieve approximate performance with merely one natural language prompt rather than complicated demonstrations, which has been proven as a simple and efficient paradigm [2]. Numerous efforts have been thrown into searching for better prompts that can benefit zero-shot CoT reasoning. Plan-and-Solve [6] employs a human-crafted prompt to break down the

---

† Work done during an internship at Alibaba Cloud Computing   ‡ Project lead
* Corresponding author

38th Conference on Neural Information Processing Systems (NeurIPS 2024).

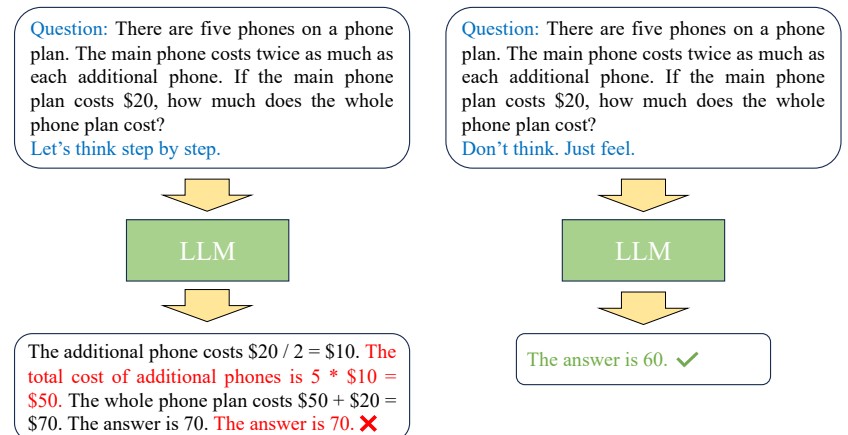

Figure 1: Input the same question with two different prompts to guide the LLM to answer it. Blue words are format tokens and prompts, red words mark wrong reasoning steps.

question and automatically generates reasoning steps. OPRO [7] takes the LLM as an optimizer to update a zero-shot CoT prompt iteratively and produce corresponding optimized prompts for a given task. Self-discover [9] selects relevant atomic reasoning modules (e.g. breaking down problems, critical thinking) for a given task, then adapts and customizes those modules to fit the task.

All prior methods focused on constructing prompts from the task perspective, aiming to find the optimal task-level prompt. Seeking the optimal prompt for a given task may achieve compelling performance, beating other prompts on the dataset scale. However, from the perspective of instance, the task-level optimal prompt within a dataset may have adverse effects on certain instances, whereby the model, capable of correctly answering them under other sub-optimal task-level prompts [11–16]. Figure 1 illustrates an instance from GSM8k dataset [17], this is a simple question that can be straightforwardly answered correctly under "Don't think. Just feel.", which is generally regarded as a less favorable prompt, but "Let's think step by step" guides the LLM to bad reasoning in some steps. Therefore, an instance-wise zero-shot CoT prompt is more plausible for better reasoning and may achieve a cap-breaking performance compared to the task-level optimal prompt.

Nevertheless, the severe challenge of choosing one of the suitable prompts for each instance remains: the difficulty of understanding why some reasoning processes succeed while others fail. To meet such a challenge, we intend to detect the mechanism of zero-shot CoT which is an unclear mystery [18–21]. Neuron saliency score analysis is an important approach for observing the information flow during the model inference [22–25], by which we can observe a click of the dynamic reasoning process in certain steps. After comprehensive investigation across several LLMs and tasks, we find that a successful reasoning procedure tends to satisfy the following conditions: the semantic information of the question should be aggregated to the prompt first, and the reasoning steps gather information from both the original question and the synthesized question-prompt semantic information. Otherwise, it is more likely to be a failure reasoning. Such a saliency score phenomenon is in line with human intuition, as the question is the beginning of reasoning, one needs to understand it first, then solve it following the rules within the prompt while always concerning the question itself.

Inspired by the above findings, we further propose an instance-adaptive prompting strategy (IAP) for zero-shot CoT reasoning. Given a list of prompts in distinct styles, we try to recognize good ones that elicit LLMs to reason toward the correct answer while avoiding bad CoT reasoning, referring to the analytical results. We conduct comprehensive experiments with IAP and existing methods with multiple LLMs on various tasks. Experimental results show that the IAP can consistently improve the overall performance of LLMs such as LLaMA-2-13B-Chat, LLaMA-3-8B-Instruct, and Qwen-14B-Chat on kinds of reasoning tasks including math, logic, and commonsense reasoning. Specifically, the IAP strategy achieves a 2%-4% accuracy enhancement across tasks and models compared to the optimal task-level prompt. Our contributions can be summarized as follows:

- We look into the inside interactions among three components (*i.e.*, question, prompt, rationale) in zero-shot CoT reasoning through the saliency score analysis and discover that good reasoning rationale tends to aggregate information from both the question and the prompt,

in which the prompt first gathers information from the question. In contrast, bad reasoning probably ignores one of them.

- We propose the IAP – an instance-level adaptive prompting strategy based on our findings to achieve better CoT reasoning by selecting a proper prompt that can elicit LLMs to reason from some given prompts for each question correctly.

- Extensive experiments illustrate the superior performance of our instance-level adaptive prompting zero-shot CoT strategy, demonstrating the effectiveness of our findings for differentiating the reasoning processes with saliency scores.

## 2 Information Flow Analysis on Zero-shot CoT

It is critical to determine the key factors for good zero-shot CoT reasoning, therefore we dive into the LLMs inference process in disparate parts. There are three main components in zero-shot CoT: question $q$, prompt $p$, and rationale $r$, and we need to choose a proper tool to analyze the semantic information interactions among these components. The saliency score is a common practice for analyzing the information flow in In-Context Learning [22, 24], and we intend to adapt it to CoT reasoning to observe the information flow in the zero-shot setting. The saliency matrix is computed by multiplying an attention matrix and its gradient for the target output element-wise as follows:

$$I^{(\ell,h)} = \left| A^{(\ell,h)} \odot \frac{\partial \mathcal{L}(x)}{\partial A^{(\ell,h)}} \right| \tag{1}$$

where $x$ is the input of the model, $A^{(\ell,h)}$ represents the value of the attention matrix of the $h$-th head in the $\ell$-th layer, $\odot$ represents the operation of element-wise multiplication, and $\mathcal{L}(\cdot)$ is the loss function, which is the cross-entropy in our implementation. Since the attention module involves interactions among the whole sequence, as a view of information flow, we can compute the saliency scores between dispersed parts during zero-shot CoT reasoning.

### 2.1 Preliminary analysis

We define the reasoning that produces the right answer to a given question as good reasoning, otherwise as bad reasoning. Given various information interactions happen among reasoning steps during the model inference, choosing which step (*i.e.*, output token) to explore is critical. Despite the most popular practice being the last step for the In-Context Learning [22, 24], there are distinct circumstances in the CoT reasoning [25], our investigation shows that not all the final answers appear at the reasoning last step. To eliminate the effect of distinct LLM generation styles as much as possible, we adopt uniformly the answer generation step for all tasks as our observation time, concretely, we implement that with several regular expressions to recognize the answer step during model inference. More details are in Appendix A.2.

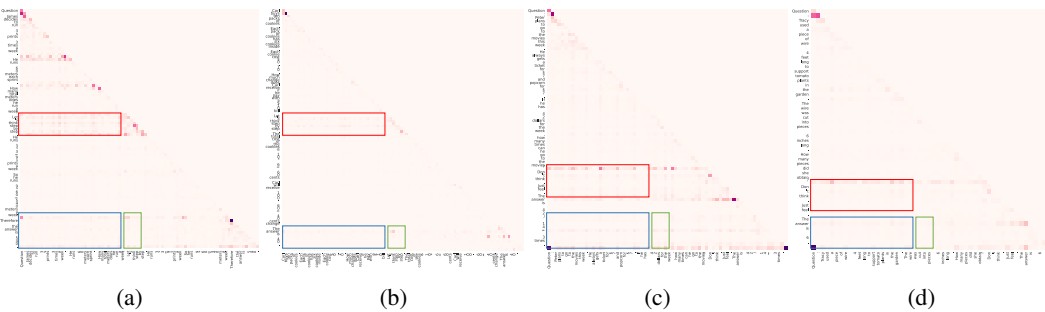

|       |       |       |       |
|-------|-------|-------|-------|
| (a)   | (b)   | (c)   | (d)   |

Figure 2: The visualization comparison of the saliency matrices between good and bad reasoning instances with two prompts, the darker the color of the pixel point in the image represents a larger saliency score. (a) and (b) are good and bad reasoning instances under "Let's think step by step.", and so as (c) and (d) under "Don't think. Just feel.", respectively. The red, blue, and green boxes in each subfigure depict the question-to-prompt, question-to-rationale, and prompt-to-rationale information flow, respectively.

Since "Let's think step by step." and "Don't think. Just feel." are two representative good and bad zero-shot prompts on task-level in [2], we select them as our test prompts. We explore the saliency score with Qwen-14B [26] on GSM8k [17] and maintain consistency in the following analysis, and we also put similar visual analysis on other models and datasets in the Appendix. To inspect saliency scores of good reasonings and bad ones, we randomly pick two pairs of good-bad reasoning instances under two prompts and visualize the saliency scores inside them in Figure 2, each subfigure depicts the mean of saliency matrices of all layers and all heads, *i.e.*, $I = \frac{1}{LH} \sum_{\ell=1}^{L} \sum_{h=1}^{H} I^{(\ell,h)}$ where $L$ and $H$ are the numbers of layer and head. As mentioned earlier, we emphasize the question, prompt, and rationale during reasoning. In Figure 2a, tokens from the first to the last of the prompt collect information from the question tokens evidently, and some tokens especially those near the answer in the rationale aggregate information from the question and prompt tokens evidently, either. In Figure 2b, things start to change, it seems that the prompt tokens fail to gather information from question tokens, not sufficiently at least, and tokens in the rationale are unable to gain much information from the question or the prompt.

The good and bad reasoning patterns under "Don't think. Just feel." are in line with the ones under "Let's think step by step.", which is shown in Figure 2c and 2d. Figure 2c illustrates that even such a prompt may guide LLMs to output the answer in very few steps after the question, the prompt tokens still capture information from the question plainly, and the limited rationale tokens proactively take advantage of information from both the question and prompt. The phenomenon in a few cases cannot illustrate any universal pattern, hence, we randomly sample 100 instances including an even number of good and bad ones to test the suitability in a larger scope. Figure 3 elaborates that good reasonings have higher mean values on the question-to-prompt, question-to-rationale, and prompt-to-rationale than those bad, justifying the above phenomenon in a broader context. We can conclude that: **For prompts that enable LLMs to reason correctly, there are significant saliency scores in the question-to-prompt and pronounced saliency scores from the question and prompt to the rationale; In contrast, for the prompts that do not lead LLMs to reason correctly, the saliency scores from the question to the prompt are usually not significant, or the flow from the question and the prompt to the rationale is not substantial.** These findings align with the human cognitive process: given a question, one needs to comprehend it first, and then address it by applying the guidelines provided in the prompt while always concerning the question itself.

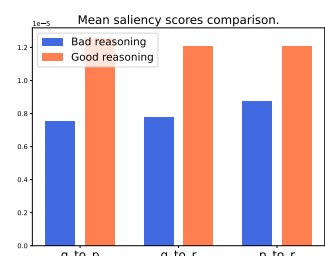

Figure 3: Comparison between mean values of randomly sampled 50 good and bad instances from GSM8k in question-to-prompt, question-to-rationale, and prompt-to-rationale.

With the saliency scores phenomenon during zero-shot CoT reasoning, we believe the strength of saliency scores among them may affect LLMs' reasoning quality. Hence, we obtain the saliency scores among the question, prompt, and CoT rationale:

$$I_{qp}^{(\ell,h)} = \frac{\sum_{(i,j) \in C_{qp}} I^{(\ell,h)}(i,j)}{|C_{qp}|} \tag{2}$$

$$C_{qp} = \{(i,j) \, | \, q_s \leq i \leq q_e, \, p_s \leq j \leq p_e\} \tag{3}$$

where $I^{(\ell,h)}(i,j)$ represents the intensity of information flow from the $i$-th token to the $j$-th token in the $h$-th head of $\ell$-th attention layer, $|C_{qp}|$ denotes the number of interactions among question tokens and prompt tokens, $q_s$ and $p_s$ are the start tokens the question and the prompt, respectively, and $q_e$ and $p_e$ are the end tokens. The saliency score of the question-to-rationale $I_{qr}^{(\ell,h)}$ and the prompt to the rationale $I_{pr}^{(\ell,h)}$ share the same computing process, only with alteration of start and end tokens.

## 2.2 Layer analysis

Popular LLMs are Transformer decoder-only models of numbers of stacked layers, and these decoder blocks play distinct roles in processing information during model reasoning. To determine the discrepancy between good reasoning and bad, we intend to check layer-wise saliency scores. In Figure 4, we visualize the saliency scores within the LLM as it processes input through its multiple

layers, and each sub-figure depicts the mean of the saliency scores of all heads in a certain layer, *i.e.*, $I_{qp}^{(\ell)} = \frac{1}{H} \sum_{h=1}^{H} I_{qp}^{(\ell,h)}$, $\ell = 1, \ldots, L$. $I_{qr}^{(\ell)}$ and $I_{pr}^{(\ell)}$ follows the same principle. The sub-figures depict saliency scores that indicate the semantic information transfer between different components of the input: the question to the prompt, the question to rationale, and the prompt to rationale. The saliency scores here serve as a quantified metric to display how the semantics of the given question and the provided prompt contribute to a well-articulated rationale. Good and bad prompts' representation across the layers enables a deeper understanding of the internal dynamics and the efficacy with which the model synthesizes input information.

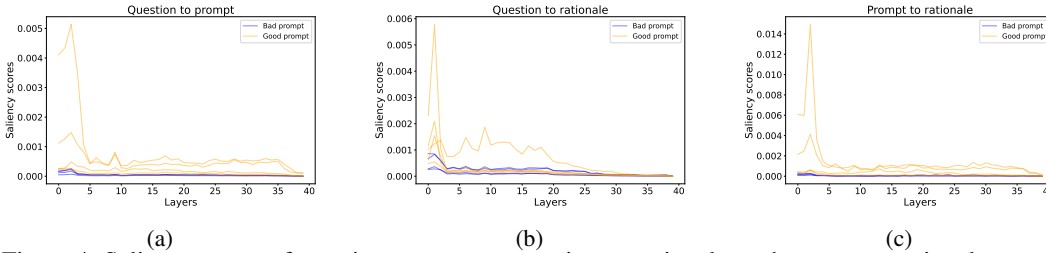

(a)  (b)  (c)

Figure 4: Saliency scores of question-to-prompt, question-to-rationale, and prompt-to-rationale across layers. The yellow lines represent prompts that effectively guide the LLMs to generate the correct answer, indicating good prompts. Conversely, the blue lines denote ineffective prompts.

As observed in Figure 4a, there is a pronounced peak in shallow layers of the LLM, demonstrating a substantial transfer of semantic content from the question to prompt in the good reasoning. This trend suggests that when the model formulates a robust prompt, it effectively aggregates the critical aspects of the original question at the outset, setting a strong foundation for later steps. Figure 4b maintains lower, yet consistent, saliency scores through the majority layers for both good and bad prompts when transferring information from the question to the rationale. This implies that while the question's semantics are integral to crafting the rationale, the direct influence is far less than the initial aggregation seen in question to prompt saliency scores. Figure 4c depicts the information flow from the prompt to the rationale, we observe a minor but stable ascending trend for good prompts. This gradual integration underscores the importance of the prompt in orchestrating the connection between the given question and rationale, particularly in the later stages process within the LLM.

Through a layer-wise analysis, we notice that **the question's information first aggregates to the prompt in shallow layers, which suggests that an appropriate prompt acts as a catalyst, enhancing the model's ability to integrate and leverage the question's meaning. Subsequently, the reasoning gathers and refines information from both the original question and the synthesized question-prompt semantics, culminating in a coherent and contextually informed rationale**. These information aggregation phenomena signify that shallow layers of the model are capable of encoding the semantic information of the question and prompt. The insights drawn from these findings evoke the potential for interpretability and reasoning capabilities of LLMs, indicating that the judicious formation of prompts can orchestrate the saliency scores in ways that affect rationales' quality.

## 2.3 Head analysis

Multi-head attention is the fundamental component in the Transformer decoder to learn the same sequence from multi-view, like different positions of Transformer blocks, scattered heads are sensitive to their locations. Figure 5 provides an in-depth examination of the instance-level saliency scores within the attention mechanism of the LLM. Representation of saliency scores as heatmap visualizations offers a detailed perspective on how semantics flow the question, prompt, and rationale propagates through individual attention heads across various layers.

Figure 5a highlights the saliency scores from the question to the prompt, attention heads at the front of the middle and end positions effectively concentrate question semantics and aid their embedding into the prompt context. Notably, this pattern corroborates our understanding that certain heads are specialized in aggregating the shallow layers, which is essential for formulating coherent prompts. Figure 5b shows the transition from the question to the rationale, reflecting the model's nuanced strategy of parsing the question to spawn a rationale and this aligns with the layer-level analysis, asserting the importance of inheriting question semantics, albeit less evidently than the question-to-

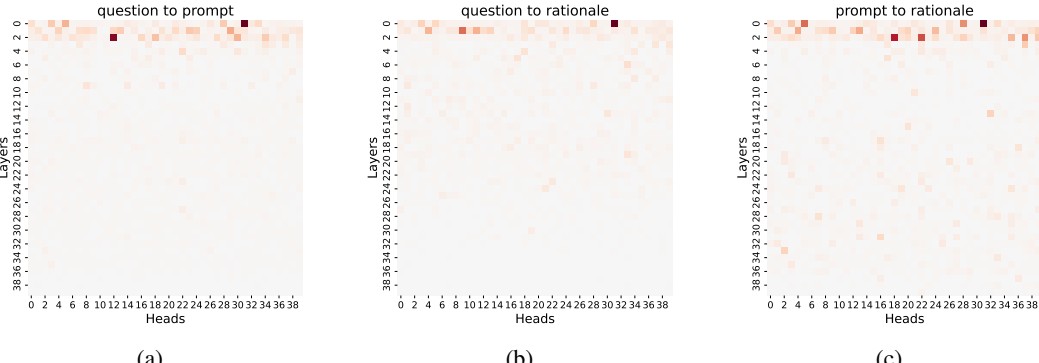

Figure 5: Saliency scores from question to prompt, question to rationale, and prompt to rationale. The color intensity across the heatmap denotes varying degrees of engagement among heads, the darker color denotes the higher score.

prompt transition. Figure 5c delineates the flow from the prompt to the rationale, scattered saliency scores across heads denote that while all heads partake in the progression towards a rationale, middle and behind heads are pivotal in harmonizing the prompt with the rationale context. This staged intertwining of prompt and rationale semantics accentuates the sophisticated nature of information assimilation in the latter reasoning.

Combining the above 3 types of head-wise analysis, we note that the distribution and intensity of attention across heads are not homogeneous but are rather intricately patterned to orchestrate a hierarchical and systematic progression of semantics. **The saliency scores from the question to the prompt and rationale are proven to be the core in the early phase, setting a solid foundation for rational derivation. The subsequent interactions that spawn the rationale further underscore the nuanced employment of attention heads in synthesizing compounds of the initial question with emergent prompt semantics**. The intelligence encapsulated in this fine-grained attention tracing elucidates the role of discrete heads in sculpting the LLM's reasoning.

## 3 Adaptive Instance-level Zero-shot CoT

To discover the zero-shot CoT reasoning capabilities of LLMs, we meticulously decompose the saliency score inherent in layers and heads to discern patterns of good and bad CoT reasonings in a fine-grained manner. In Section 2.2 and 2.3, we find that good reasonings always have higher saliency scores than bad ones in the question-to-prompt, question-to-rationale, and prompt-to-rationale, we further discover the front heads of the middle and end positions in shallow layers contribute to the saliency scores in both the question-to-prompt and question-to-rationale.

We first compute the saliency scores of question-to-prompt, question-to-rationale, as well as prompt-to-rationale. Then, for each question and a certain prompt, we compute the synthesized saliency score as follows:

$$S = \frac{1}{|\mathfrak{L}| \cdot |\mathfrak{H}|} \sum_{\ell, h \in \mathfrak{L} \times \mathfrak{H}} \lambda_1 \cdot I_{qp}^{(\ell,h)} + \lambda_2 \cdot I_{qr}^{(\ell,h)} + \lambda_3 \cdot I_{pr}^{(\ell,h)} \tag{4}$$

where $I_{qp}^{(\ell,h)}$, $I_{qr}^{(\ell,h)}$, and $I_{pr}^{(\ell,h)}$ are the question-to-prompt, question-to-rational and prompt-to-rationale saliency scores computed as Eq. 2, $\mathfrak{L}$, $\mathfrak{H}$ are the indices set of the selected layers and heads, $\mathfrak{L} \times \mathfrak{H}$ is the cartesian product of two sets, $|\mathfrak{L}|$ and $|\mathfrak{H}|$ are the number of the elements in the set, and the $\lambda_1$, $\lambda_2$, and $\lambda_3$ are hyperparameters to adjust the ratio of different saliency scores and obey $\lambda_1 + \lambda_2 + \lambda_3 = 1$. After engaging in a comparative analysis of numerous instances from various datasets with distinct LLMs and prompts, we summarize different saliency score thresholds of question-to-prompt, question-to-rationale, and prompt-to-rationale to delimit the good and bad reasonings when the inference reaches the answer step. Inspired by these analytical findings, we present a novel **I**nstance-**A**daptive **P**rompting strategy, dubbed IAP, which leverages the qualitative and quantitive saliency scores to tailor the zero-shot CoT prompting process instance-wise, thereby enhancing LLMs' reasoning ability. Our IAP framework can be instantiated through two distinct methodologies: Sequential Substitution (IAP-ss) and Majority Vote (IAP-mv).

**Sequential Substitution (IAP-ss)**   Based on the above findings, we believe that a prompt with saliency scores surpassing the corresponding threshold is considered a good prompt for a given question, consequently mitigating the need to explore further prompts. Given the training data, we can search for an appropriate threshold. This process terminates upon either identifying an optimal prompt or traversing all candidates.

**Majority Vote (IAP-mv)**   Alternatively, the IAP-mv necessitates the computation in Eq. 4 across all candidate prompts, then preserves the top maximum scores, predominant answer among these top scores is selected as the final answer. This synergistic combination ensures that the chosen prompt not only aligns with the LLM's inherent reasoning pattern but also complies with the collective intelligence inferred from an assorted selection of potential prompts.

Both methods have pros and cons: IAP-ss possesses the efficiency of a heuristic-based sequential evaluation, which needs less computational resource; while IAP-mv owns the robustness supported by the consensus-based vote. Correspondingly, IAP-ss can be constrained in its performance potency since a few irregular instances may depart from our findings; though IAP-mv may achieve better performance, it demands the comprehensive evaluation of all candidate prompts. In summary, the IAP contributes a novel perspective on the paradigm of instance-level prompting strategies that drive the frontier of zero-shot CoT reasoning with LLMs.

# 4   Experiments

## 4.1   Implementation

**Models.**   We test IAP and comparison methods on LLaMA-3-8B-Instruct [27], LLaMA-3-70B-Instruct [], LLaMA-2-13B-Chat [28], and Qwen-14B-Chat [26] since they are popular Transformer decoder-only LLMs, which is convenient for exploiting and analyzing inside architectures. For the IAP-ss, we obtain threshold values w.r.t distinct LLMs on different datasets, we compute the overall synthesized scores as defined in Eq 4. We set the generation mode to *greedy-decoding* to minimize irrelevant confounders during the model inference to ensure the answers to fixed questions under the same model and prompt, and all the experiments are run on an 8x NVIDIA A100 GPU server.

**Baselines. Answer majority vote (AMV)** is a simple method implemented by choosing the most popular result of all prompt candidates for a given question as its final answer. **OPRO** [7] takes the LLM as an optimizer to update a zeros-hot CoT prompt iteratively and produce corresponding optimized prompts for copious tasks. **Self-Discover** [9] selects relevant atomic reasoning modules (*e.g.*, decomposing problems, critical thinking) for a given task, then adapts and customizes those modules to fit the task. These two frameworks aim to search for an appropriate prompt, similar to our purpose. We choose them as comparisons with the IAP to observe the performance difference between instance-level and task-level zero-shot CoT prompting.

**Tasks & Metrics.** GSM8k [17] is a challenging dataset for assessing the capability of language models in multi-step math reasoning. SVAMP [29] is presented for one-step math reasoning, which is easier than GSM8k. CommonsenseQA [30] is designed to evaluate a model's capacity for commonsense reasoning with questions that demand commonsense knowledge. The MMLU [31] can assess a model's multi-task learning abilities across natural language inference, commonsense reasoning, question answering, *etc*. Causal Judgement and Tracking Shuffled Objects are two sub-tasks in BBH [32], the former specifically tests a model's ability to reason about the dynamics and interactions of objects in a given scenario and the latter presents scenarios that require identifying the underlying causes and effects of specific events or phenomena. We select the GSM8k and SVAMP for math reasoning, Causal Judgement, and Tracking Shuffled Objects-5-Objects for logic reasoning, CSQA, and MMLU for Commonsense reasoning. For all tasks, we adopt Accuracy as the only evaluation metric.

**Zero-shot CoT Prompts.** In the following part, we use #1 to represent "Let's think step by step.", #2 denotes "First,", #3 is "The answer is after the proof.", #4 is "Before we dive into the answer,", #5 is "Let's solve this problem by splitting it into steps.", #6 is "Let's think about this logically.", #7 is "It's a beautiful day.", #8 is "Don't think. Just feel.", and #9 is "By the fact that the earth is round," and we implement the IAP by enabling these 9 prompts as the candidates.

Table 1: Zero-shot CoT results with **LLaMA-3-8B-Instruct** and **Qwen-14B-Chat** under various prompts, the results of **LLaMA-3-70B-Instruct** and **LLaMA-2-13B-Chat** is in Appendix A.1. Each column stands for a group of task categories, T-Obj. is for Tracking Shuffled Objects which are from the BBH. The "Optimizer-generated prompt" refers to the prompts for each task generated with the algorithm in [7].

| Prompt | Math | | | | Logic | | | | Commonsense | | | |
|---|---|---|---|---|---|---|---|---|---|---|---|---|
| | GSM8k | | SVAMP | | C-Judge. | | T-Obj. | | CSQA | | MMLU | |
| | LLaMA3 | Qwen | LLaMA3 | Qwen | LLaMA3 | Qwen | LLaMA3 | Qwen | LLaMA3 | Qwen | LLaMA3 | Qwen |
| #1 | 64.52 | 58.00 | 73.67 | 66.00 | 4.28 | 9.09 | 40.00 | 13.20 | 59.87 | 54.63 | 55.79 | 42.48 |
| #2 | 57.54 | 52.01 | 67.00 | 51.67 | 14.97 | 17.11 | 29.60 | 16.80 | 64.95 | 49.06 | 50.35 | 61.93 |
| #3 | 41.62 | 60.50 | 62.00 | 67.33 | 12.30 | 9.63 | 12.40 | 23.20 | 59.62 | 48.73 | 43.51 | 48.25 |
| #4 | 58.98 | 57.47 | 60.33 | 72.00 | 13.90 | 6.95 | 24.40 | 15.60 | 64.95 | 36.61 | 48.95 | 74.21 |
| #5 | 56.25 | 55.50 | 57.33 | 60.67 | 5.35 | 28.34 | 20.00 | 16.00 | 55.28 | 41.20 | 46.67 | 76.84 |
| #6 | 62.74 | 58.07 | 76.00 | 71.33 | 3.74 | 3.21 | 24.40 | 17.60 | 59.87 | 63.23 | 56.67 | 55.25 |
| #7 | 61.79 | 27.82 | 66.67 | 42.67 | 2.14 | 1.07 | 24.00 | 16.80 | 33.25 | 23.42 | 42.28 | 57.19 |
| #8 | 31.69 | 26.25 | 57.00 | 57.67 | 16.04 | 1.07 | 16.80 | 2.00 | 35.71 | 34.56 | 26.32 | 9.30 |
| #9 | 12.05 | 20.39 | 39.67 | 21.00 | 2.67 | 2.14 | 13.60 | 10.80 | 50.61 | 61.75 | 20.18 | 30.70 |
| AMV (all) | 52.54 | 28.22 | 74.33 | 51.33 | 17.06 | 26.10 | 12.60 | 1.44 | 62.41 | 46.52 | 52.53 | 52.46 |
| AMV (#1-7) | 57.82 | 57.98 | 77.00 | 55.33 | 18.13 | 27.50 | 20.80 | 8.80 | 65.03 | 57.70 | 41.23 | 63.86 |
| OPRO | 65.96 | 36.01 | - | - | 18.18 | 19.79 | 28.00 | 4.00 | - | - | - | - |
| Self-dis | 8.50 | 56.33 | 15.33 | 52.67 | 10.70 | 11.23 | 36.00 | 24.00 | 60.03 | 57.33 | 37.37 | 52.63 |
| IAP-ss | 65.36 | 61.57 | 75.33 | 71.67 | 16.57 | 26.74 | 38.80 | 24.00 | 65.68 | 64.37 | 56.49 | 77.07 |
| IAP-mv | **66.34** | **62.81** | **77.33** | **73.33** | **19.25** | **29.95** | **42.40** | **25.60** | **68.39** | **65.68** | **59.65** | **78.95** |

## 4.2 Results

Prompts steer these LLMs to achieve different results in multiple tasks, and no single prompt can get an overwhelming performance on all datasets, which makes our research on the mechanism of zero-shot CoT valuable. Table 1 shows the zero-shot CoT reasoning results with LLaMA-3-8B-Instruct and Qwen-14B-Chat and various prompts on 3 reasoning tasks, we put the results of LLaMA-2-13B-Chat and larger LLaMA-3 70B in the Appendix since LLaMA models share a quite similar architecture.

**Math reasoning.** Compared with these prompts, IAP-mv improves the LLaMA-3 and Qwen's accuracy on GSM8k from 64.52%, 60.50% to 66.34%, 62.81% respectively. On SVAMP, IAP-mv obtains a 1.33% improvement on both models compared to the task-level-optimal prompt. It is worth noting that OPRO and Self-discover are unstable with different LLMs and datasets, indicating the unstable characteristics of task-level prompting. Results on these two math reasoning datasets demonstrate the IAP can benefit the math reasoning task.

**Logic reasoning.** For Causal Judgement, IAP-ss and IAP-mv enhance the accuracy of the task-level optimal prompt and IAP-mv outperforms OPRO, which is optimized by numerous iterations. For Tracking shuffle Objects, IAP-mv performs well with Qwen while achieving a sub-optimal accuracy with LLaMA-3, IAP-mv still obtains strong results, improving 2.4% and 2.3% with LLaMA-3 and Qwen, separately.

**Commonsense reasoning.** On CSQA, the IAP improves the accuracy of the former best for 3.44% with LLaMA-3, and 2.45% with Qwen. On MMLU, LLaMA-3 and Qwen obtain improvement to a large margin, either. We note that improving IAP-mv and IAP-ss on commonsense reasoning is more salient than the other two reasoning tasks, demonstrating the effectiveness of the saliency score-based prompting strategies.

Table 2: Accuracy (%) of Consistency and Complementary prompts with **IAP-mv** on 3 tasks with **LLaMA-3-8B-Instruct** and **Qwen-14B-Chat**. The results of **LLaMA-2-13B-Chat** are at Appendix A.1.

| | Math | | | | Logic | | | | Commonsense | | | |
|---|---|---|---|---|---|---|---|---|---|---|---|---|
| | GSM8k | | SVAMP | | C-Judge. | | T-Obj. | | CSQA | | MMLU | |
| | LLaMA3 | Qwen | LLaMA3 | Qwen | LLaMA3 | Qwen | LLaMA3 | Qwen | LLaMA3 | Qwen | LLaMA3 | Qwen |
| Instr. | 65.05 | 61.18 | 76.33 | 72.67 | 17.11 | 29.41 | 41.60 | 24.80 | 67.57 | 64.54 | 57.89 | 78.25 |
| Misl. | 31.84 | 27.37 | 57.67 | 59.00 | 16.04 | 2.14 | 17.20 | 10.40 | 51.27 | 63.14 | 26.84 | 31.05 |
| Instr.+Irr. | 65.35 | 61.49 | 76.67 | 73.00 | 18.18 | 28.34 | 42.00 | 24.00 | 67.24 | 64.21 | 58.77 | 78.42 |
| Misl.+Irr. | 62.55 | 28.13 | 67.33 | 59.33 | 16.04 | 2.14 | 24.80 | 11.20 | 52.83 | 62.41 | 44.56 | 58.95 |
| Instr.+Misl. | 64.90 | 61.41 | 77.00 | 72.67 | 18.72 | 2.14 | 41.60 | 23.60 | 52.17 | 62.49 | 57.54 | 78.24 |

Apart from the above comparison, we can also observe that the answer majority vote among all prompts performs poorly in some tasks, in contrast, AMV (#1-7) can enhance it by eliminating misleading prompts. Such results indicate the instability of the AMV, whose performance can be affected by prompt candidates. Our IAP-mv outperforms it by a large margin, demonstrating that most prompts can lead the LLM to generate wrong answers for a given question, reaching only a few correct answers, i.e., such methods cannot recognize good or bad reasoning. Our IAP-mv can handle that with the analysis for information flow in reasoning, i.e., IAP-mv can differentiate good and bad reasoning, validating the effectiveness of our proposed strategy.

### 4.3 Ablation Studies

**Consistency & Complementary**  The success of zero-shot prompting for CoT reasoning lies in the semantic information within those prompts, when the LLM receives a prompt, it would generate rationales by obeying the meaning of the prompt as much as possible. According to semantics, [2] categorizes these zero-shot CoT prompts into 3 types: instructive, misleading, and irrelevant, and we further define that prompts in the same category are consistent, or otherwise they are complementary. To detect which type of prompt combination contributed to the performance, we divide the 9 prompts into 3 consistency groups, but the irrelevant group contains only one prompt, thus we evaluate the complementary on the other two. For the complementary groups, we build them two-by-two. Table 2 depicts the performance of each group. We employ IAP-mv since it manifests a stronger capability in harnessing multiple prompts. we can observe that each pair of combinations can improve the performance, and instructive and

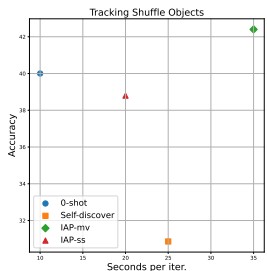

Figure 6: Efficiency comparison with LLaMA-3-8B-Instruct on the Tracking Shuffle Objects, 0-shot denotes the best task-level prompt.

irrelevant combinations achieve better outcomes than others, which comes from the base performance of instructive prompts.

**Efficacy**  The order and number of prompt candidates are critical for the accuracy and efficacy of IAP-ss, in this paper, we adopt the #1-9 order to conduct IAP, and we also tried other settings. In Table 3., #9 is the worst task-level prompt, and #6 is the best task-level prompt, achieving the highest accuracy among all the prompt candidates while consuming the least time.

The prompt order of the 3rd row is accuracy-decreased on SVAMP, the 4th row is accuracy-increased, and the last row is our default setting, which obtains the best performance. This table shows that IAP-ss can cost less time with fewer prompt candidates but may obtain limited results, however, even fewer improper candidates could take a lot of computing time. Therefore, the time cost of IAP-ss is not a major issue if prompt candidates are in an appropriate order. As we mentioned in Section 3, the IAP-mv trades efficiency for performance, and IAP-ss emphasizes efficiency. We introduce the reasoning time (seconds) for

Table 3: Accuracy and inference time (s) with different prompt orders and numbers of LLaMA-3-8B-Instruct on SVAMP.

| Order | Acc | Time |
|---|---|---|
| #9 | 39.67 | 2860 |
| #6 | 76.00 | **2657** |
| #9, 8, 5, 4, 3 | 63.66 | 3870 |
| #6, 1, 2, 7, 3 | 76.66 | 5216 |
| #1, 2, 3, 4, 5, 6, 7, 8, 9 | **77.33** | 4751 |

each iteration complete as the metric to measure the efficiency and conduct time-consuming experiments under the same setting to show the cost of IAP-mv, IAP-ss, and Self-Discover on the Tracking Shuffle Objects dataset, results are shown in Figure 6. All these strategies increase the computation cost to a certain degree, while IAP-ss may bring accuracy decreases than the task-level optimal prompt, it beats Self-discover. Though IAP-mv is the most time-consuming, it can improve performance, therefore, the two IAP strategies can be employed as trade-offs in different demand prioritization applications.

## 5 Related Work

CoT reasoning [1] advances the reasoning abilities of LLMs by demonstrating a series of logical steps preceding the input demonstration. Building on the groundwork laid by CoT, Self-consistency [3] innovates through a margin decoding strategy that emphasizes the majority paths to derive the final

answer, presenting a significant leap in CoT reasoning. Similarly, the Least-to-most [33] strategy decomposes a complex question into manageable subquestions, addressing them progressively to achieve a comprehensive solution. Furthermore, the Plan-and-Solve [6] automates the generation of reasoning steps through a meticulously crafted prompt, streamlining the breakdown of questions into digestible parts that can be tackled sequentially.

Promisingly, the AutoHint framework [34] augments the original prompt with enriched instructions extracted from contextual demonstrations. Similarly, the COSP [35] capitalizes on answer pools derived from training sets to compute outcome entropy, inspired by the notion of self-consistency, thereby refining the selection process for QA pairs used during test set demonstrations. In specialized prompting, MathPrompter [36] specifically caters to mathematics problems, employing handcrafted prompts to generate diverse algebraic expressions or Python functions. In contrast, Progressive-Hint Prompting [37] facilitates dynamic interactions between users and LLMs, guiding the reasoning with hints to generate from previous answers. Moreover, InstructZero [38] leverages an open-source LLM to enhance soft prompts relevant to Bayesian tasks, iteratively optimizing prompts to navigate through complex reasoning landscapes.

Advanced prompting approaches such as SelfzCoT [39] and Meta-prompting [40] showcase the evolutionary trajectory of prompting, which generates semantic and code prompts through a root prompt to obtain precise answers, while Meta-prompting deconstructs complex tasks into simpler sub-tasks, each addressed by specialized models to foster inter-model communication and apply intricate reasoning. Lastly, methodologies like OPRO [7] and the innovative concept of evolutionary prompting [16] aim to recursively optimize CoT prompts and generate varied prompts through mutations and crossovers. Self-discover [9] selects relevant atomic reasoning modules (*e.g.*, breaking down problems, critical thinking) for a given task, then adapts and customizes those modules to fit the task. Implement the customized reasoning structure when solving task instances. These workarounds significantly contribute to developing zero-shot CoT prompts that guide LLMs toward more accurate problem framing, intermediate reasoning, and final answers.

## 6    Conclusion

In this paper, we aim to delve into the mechanism of LLMs in zero-shot CoT reasoning from the perspective of information flow to understand what happened during this process, and we find stronger saliency scores within question-to-prompt and question-to-rationale can lead to better LLM reasoning. To investigate these phenomena nuancedly, we go deep into the Transformer layers and attention heads in the LLM and find the front of the middle and final heads in shallow layers carry more information during information flows. Inspired by that, we present an instance-adaptive zero-shot prompting strategy for better CoT reasoning. To demonstrate our findings, we conduct comprehensive experiments on several LLMs and tasks, and the results show our proposed strategies can improve the performance of LLMs on all candidate prompts, highlighting our interpretation of zero-shot CoT in the view of information flow.

## Limitations

In this work, we select the answer step as the key step to investigate and visualize the saliency scores, even in most instances it can be located well, and some irregular answers can not be identified precisely, such a factor may affect the generality and accuracy of our analysis. Different LLMs may have distinct patterns under the zero-shot CoT reasoning, for example, our analysis and conclusion can not meet all models. Despite our research providing insight into understanding the underlying workflow of zero-shot CoT reasoning, it cannot be the only interpretation, and we believe there must be better means to explain that.

## Acknowledgement

This work is supported in part by Alibaba Research Intern Program, the National Natural Science Foundation of China (No.62272191, No,62402440), the International Science and Technology Cooperation Program of Jilin Province (No. 20240402067GH), and the Science and Technology Development Program of Jilin Province (No. 20220201153GX).

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

# A Appendix

## A.1 IAP & Baseline Experiments on LLaMA-2-13B-Chat and LLaMA-3 70B

Table 4 and Table 6 are supplementary for Table 1 and Table 2, the results here basically are coincidence with the Experiment section in the main body. Table 5 shows the results of single prompts and IAP-mv to demonstrate the generality for a large LLM. However, the IAP did not obtain salient enhancement, which may caused by the irregular output formats of LLaMA-2.

Table 4: Zero-shot CoT results with **LLaMA-2-13B-Chat** under various prompts and other baselines.

| Zero-shot CoT Prompt | Math | | Logic | | Commonsense | |
| --- | --- | --- | --- | --- | --- | --- |
| | GSM8k | SVAMP | C-Judge. | T-Obj. | CSQA | MMLU |
| #1 | 30.86 | 37.33 | 11.76 | 5.60 | 31.29 | 37.54 |
| #2 | 32.90 | 43.67 | 13.90 | 8.00 | 32.02 | 42.81 |
| #3 | 23.20 | 40.33 | 24.06 | 4.80 | 38.08 | 41.23 |
| #4 | 29.34 | 36.33 | 16.58 | 1.60 | 27.44 | 47.89 |
| #5 | 29.19 | 41.67 | 14.97 | 0.80 | 43.41 | 39.30 |
| #6 | 30.93 | 41.33 | 32.68 | 9.20 | 44.06 | 26.14 |
| #7 | 19.94 | 36.67 | 14.97 | 2.80 | 4.50 | 55.61 |
| #8 | 14.03 | 42.33 | 7.49 | 3.20 | 1.72 | 57.37 |
| #9 | 18.50 | 45.67 | 9.62 | 1.60 | 37.67 | 28.77 |
| OPRO | **33.66** | - | 13.37 | 0.08 | - | - |
| Self-disc | 7.43 | 17.33 | 10.16 | 2.40 | 33.01 | 58.07 |
| IAP-ss | 31.35 | 45.36 | 32.56 | 8.80 | 44.55 | 57.72 |
| IAP-mv | 32.78 | **47.15** | **33.47** | **9.60** | **45.76** | **58.25** |

Table 5: Zero-shot CoT results with **LLaMA-3 70B** under various prompts.

| Zero-shot CoT Prompt | Math | | Logic | | Commonsense | |
| --- | --- | --- | --- | --- | --- | --- |
| | GSM8k | SVAMP | C-Judge. | T-Obj. | CSQA | MMLU |
| #1 | 87.79 | 82.33 | 38.50 | 12.40 | 67.73 | 37.02 |
| #2 | 89.16 | 86.33 | 54.55 | 30.00 | 56.10 | 50.18 |
| #3 | 81.73 | 83.33 | 49.73 | 23.20 | 55.69 | 44.56 |
| #4 | 82.64 | 84.33 | 42.25 | 60.40 | 41.36 | 52.11 |
| #5 | 82.71 | 84.00 | 36.36 | 6.80 | 61.75 | 52.63 |
| #6 | 87.79 | 82.33 | 44.39 | 16.00 | 67.73 | 35.79 |
| #7 | 81.43 | 85.67 | 47.59 | 24.00 | 29.98 | 14.56 |
| #8 | 53.53 | 75.67 | 55.61 | 18.40 | 29.24 | 22.56 |
| #9 | 51.71 | 58.33 | 44.92 | 20.40 | 36.94 | 43.33 |
| IAP-mv | **89.84** | **87.33** | **56.20** | **62.00** | **69.04** | **54.39** |

Table 6: Accuracy (%) of Consistency and Complementary prompts with **IAP-mv** on 3 tasks with **LLaMA-2-13B-Chat**.

| | Math | | Logic | | Commonsense | |
| --- | --- | --- | --- | --- | --- | --- |
| | GSM8k | SVAMP | C-Judge. | T-Obj. | CSQA | MMLU |
| Instr. | 32.22 | 44.58 | 33.15 | 9.60 | 45.21 | 48.42 |
| Misl. | 19.48 | 46.70 | 10.16 | 3.20 | 37.67 | 57.54 |
| Instr.+Irr. | 31.39 | 44.66 | 32.62 | 9.60 | 44.39 | 56.49 |
| Misl.+Irr. | 20.47 | 46.10 | 15.51 | 3.20 | 38.17 | 58.07 |
| Instr.+Misl. | 32.52 | 46.32 | 33.16 | 9.60 | 44.39 | 57.89 |

## A.2 Answer Step Recognition

We prepare 3 types of answer formats to recognize the answer step while LLMs reasoning, concretely, employs the regular expression to judge whether the model has just output the answer to the given question. Once we detect some pre-defined patterns, we break the LLM's generation for loop and compute the saliency scores at this time step. We put the recognition formats in Table 7.

Table 7: Regular expressions for answer step recognition.

| Style. | RegExp |
|---|---|
| Numbers. | (Therefore, the) answer is(:) (Arabic numerals)(,l.) |
| Choices. | (Therefore, the) (answer\|choice) is(:) (A-Za-z)(,l.) |
| Y/N. | (Therefore, the) answer is (Yes\|No)(,l.) |

## A.3 Information Flow Analysis on Other LLMs

In our investigation, we analyzed the information flow with the same method in different LLMs on various datasets and found that the phenomena of saliency scores for all LLMs on most datasets are quite similar, so we put the analysis process of Qwen-14B-Chat on GSM8k to maintain consistency in the narrative subject and present our analysis conclusion. Similarly, the head analysis results for good and bad reasoning are consistent with other LLMs or datasets.

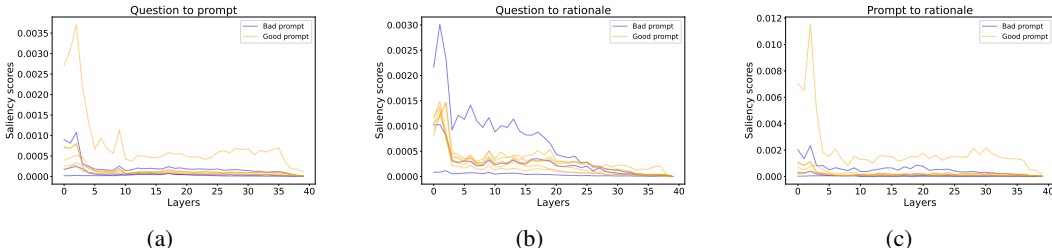

(a)                      (b)                      (c)

Figure 7: Saliency scores across layers of LLaMA-2-13B-Chat on CSQA, in contrast to Qwen-14B-Chat on GSM8k in the main text.

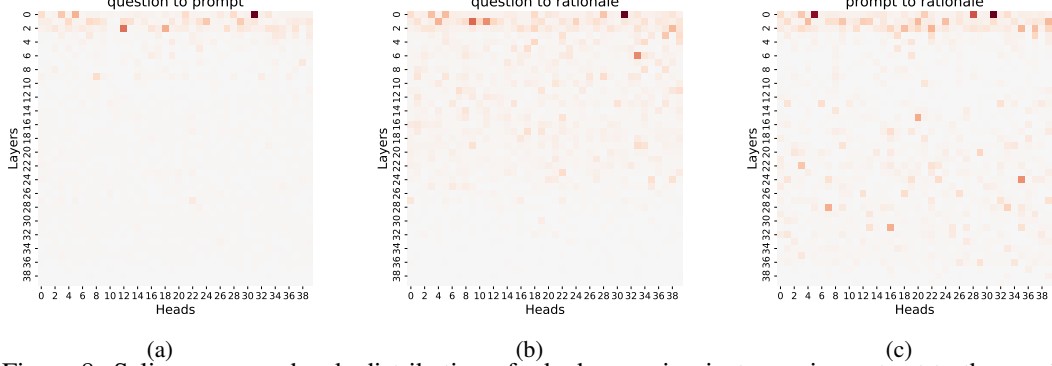

(a)                      (b)                      (c)

Figure 8: Saliency scores heads distribution of a bad reasoning instance, in contrast to the good reasoning in the main text.

## A.4 Thresholds and Majority Number

For IAP-ss, we obtain threshold values with regard to distinct LLMs on different training sets, we compute the overall synthesized scores (defined in eq (4) in Section 3) to divide up the good and bad reasoning paths and adopt the thresholds that classify reasoning well. Such as, the threshold of LLaMA-3 8B on GSM8k is 5.5e-6, and the identification of the thresholds of different LLMs on different datasets is the same and it is simple and doesn't not need much time. In practice, we consider reasoning with a value higher than the threshold as good, otherwise bad. We have tried different thresholds, and the best performance is shown in Table 8. As for the IAP-mv, we select

top-k (hyper-parameter, k=3) values and use the majority result as the final result, we also tried other k values and k=3 is the best among all other values with LLaMA-3-8B-Instruct and pick some results on 3 datasets in Table 9.

Table 8: Accuracy of different thresholds with LLaMA-3-8B-Instruct on GSM8k.

| Threshold | Acc |
|---|---|
| 7.0e-6 | 59.82 |
| 6.0e-6 | 62.77 |
| 5.0e-6 | 64.67 |
| 4.0e-6 | 62.40 |
| 5.5e-6 | 65.36 |

Table 9: Accuracy of different thresholds with LLaMA-3-8B-Instruct on GSM8k.

| K | MMLU | C-Judge | T-Obj |
|---|---|---|---|
| 1 | 52.98 | 15.51 | 36.80 |
| 5 | 55.96 | 18.72 | 40.00 |
| 3 | 59.65 | 19.25 | 42.40 |

