# OpenReview forum: "Instance-adaptive Zero-shot Chain-of-Thought Prompting"
_NeurIPS.cc/2024/Conference — NeurIPS 2024 poster_

### Official Review · Reviewer_43EH · 2024-06-25

**Soundness:** 3
**Presentation:** 3
**Contribution:** 3
**Rating:** 5
**Confidence:** 4

**Summary:**

This paper explores the inside interactions among three components(i.e., question, prompt, rationale) in zero-shot CoT reasoning through the saliency score analysis, and discover the distinct characteristics of good and bad reasoning in terms of information flow. Based on above findings, the authors propose the IAP - an instance-level adaptive prompting strategy designed to enhance CoT reasoning by selecting an appropriate prompt that can guide LLMs to reason correctly from a given set of prompts for each question. To demonstrate the effectiveness of the proposed method, the authors conducted extensive experiments on the LLaMA-3 8B, LLaMA-2 13B, and Qwen14B models across Math, Logic, and Commonsense tasks.

**Strengths:**

* **The motivation  of this paper is quite intuitive.** Through the example in Figure 1, the authors illustrate that an instance-wise zero-shot CoT prompt is more plausible for better reasoning and may achieve a cap-breaking performance compared to the task-level optimal prompt, which is easy to understand.

* **Using Neuron saliency score to analyze the information flow of the question, prompt, and rationale is innovative and promising.** The authors conduct both qualitative and quantitative analyses on good-bad reasoning instances, discovering different patterns of information flow in good-bad reasoning. They further delve into a deeper analysis from the perspectives of Layer and Head, yielding valuable conclusions.

* **Extensive experiments are conducted to validate the effectiveness of proposed method.** The authors test the proposed IAP and comparison methods on LLaMA-3 8B, LLaMA-2 13B, and Qwen14B across Math, Logic, and Commonsense reasoning tasks.

**Weaknesses:**

* **The  proposed method is not clearly explained, some details are missing.** For instance, in Sequential Substitution (IAP-ss), it is not specified what the corresponding threshold is or how it was obtained. Similarly, in Majority Vote (IAP-mv), there is no explanation provided for how many top maximum scores are preserved. The lack of details in the proposed methods can be quite confusing.

* **The experimental setup is unfair.** For the main "effective" method proposed, IAP-mv, which obtains the final results through Majority Vote (similar to self consistency), it is unreasonable to compare it with baselines that only perform inference once. The improvement brought about by this method could very well be the result of ensemble inference from multiple reasoning paths [1][2][3], rather than the strategy implemented by the author's previous good-bad reasoning findings.

* **Some experimental conclusions are missing or somewhat unreasonable.** For instance, in the experiment of Consistency & Complementary, the authors only provide the experimental setup and show the corresponding results in Table 2 without giving any conclusions or analysis. Furthermore, in the analysis experiment for Efficacy, the author demonstrated through Figure 6 that while IAP-ss also incurs additional overhead, its accuracy compared to 0-shot also declines to some extent. Therefore, the conclusion stated by the authors that "the two IAP strategies can be employed as trade-offs in different demand prioritization applications" is not in line with the facts, as IAP-ss does not provide any gain compared to the baseline 0-shot.


[1] Making Large Language Models Better Reasoners with Step-Aware Verifier

[2] Diversity of Thought Improves Reasoning Abilities of LLMs

[3] Answering Questions by Meta-Reasoning over Multiple Chains of Thought

**Questions:**

See weeknesses.

Although this paper performs quite well in the section of Information Flow Analysis on Zero-shot CoT, there are numerous flaws in its experiments that fail to convince me.  If the authors can address my concerns, I would consider raising my score.

**Limitations:**

I think the authors have addressed their limitations.

---

> ### Author Rebuttal · Authors · 2024-08-07
>
> Thank you for the constructive advice and comments, which have greatly improved our manuscript, and we are glad to reply to your invaluable suggestions and questions.
>
> ### **Response to Weaknesses**
>
> **Weakness 1:**
> The proposed method is not clearly explained, some details are missing.
>
> **Response to Weakness 1:**
> For IAP-ss, we obtained threshold values with regard to distinct LLMs on different training sets, we computed the overall synthesized scores (defined in eq (4) in Section 3) to divide up the good and bad reasoning paths and adopted the thresholds whose classify reasoning well. Such as, the threshold of LLaMA-3 8B on GSM8K is 5.5e-6, and the identification of the thresholds of different LLMs on different datasets is the same and it is simple and doesn't not need much time. In practice, we considered reasoning with a value higher than the threshold as good, otherwise bad. We have tried different thresholds, and the best performance is shown in the following table.
>
> | threshold | accuracy |
> | --- | --- |
> | 7.0e-6 | 59.82 |
> | 6.0e-6 | 62.77 |
> | 5.0e-6 | 64.67 |
> | 4.0e-6 | 62.40 |
> | 5.5e-6 | **65.36** |
>
> We can see that an improper threshold can affect the performance of the IAP-ss, whether higher or lower. It comes from that higher thresholds tend to recognize some good reasoning instances as bad ones, and lower thresholds may overlook some bad reasoning.
>
> As for the IAP-mv, we selected top-k (hyper-parameter, k=3) values and adopted the majority result as the final result, we also tried other k values and k=3 is the best among all other values with LLaMA-3 8B and picked some results on 3 datasets in the following table.
>
> | k | MMLU | C-Judge. | T-Obj |
> | --- | --- | --- | --- |
> | 1 | 52.98 | 15.51 | 36.80 |
> | 5 | 55.96 | 18.72 | 40.00 |
> | 3 | **59.65** | **19.25** | **42.40** |
>
> The k=3 achieves the best performance on most datasets, therefore, we selected the hyper-parameter k=3 as the default value in the paper. We are sorry for confusing you in this part and we have elaborated these details in the new version.
>
>
> **Weakness 2:**
> The experimental setup is unfair.
>
> **Response to Weakness 2:**
> We are sorry for not introducing the comparable approaches thoroughly in the baseline paragraph of Section 4.1, and we would like to replace the current version with the new one:
>
>    > In practice, the OPPR optimizes a meta prompt on different datasets with distinct LLMs for multiple rounds to guide LLMs to produce better prompts, requiring numerous optimization steps (i.e., inferences). As for Self-Discover, it consists of selecting, adapting, implementing, and reasoning, which also needs multi-round inference in every step.
>
> Given the above description, we think that comparing the IAP with the above methods is not unfair.
>
> The profits of IAP-mv didn't simply come from the multiple reasoning path, we computed the synthesized saliency scores (defined in eq (4)., as an application of former analysis) of all prompt candidates, and conducted a majority vote based on these synthesized scores. For a considerable proportion of questions, the LLM could be guided to generate wrong results with most prompts but only a few right results and our IAP-mv can handle it, thereby outperforming the majority final results vote. We compared the IAP-mv and direct majority results vote results in the following table:
>
> | Method | GSM8K | SVAMP | C-Judge. | T-Obj. | CSQA | MMLU |
> | ------ | ----- | ----- | -------  | ------ | ---- | ---- |
> | Majority Vote. (Qwen 14B) | 28.22 | 51.33 | 26.10 | 1.44 | 46.52 | 52.46 |
> | IAP-mv (Qwen 14B) | **62.81 (+34.59)** | **73.33 (+22.00)** | **29.95 (+3.85)** | **25.60 (+24.16)** | **65.68 (+19.16)** | **78.95 (+26.49)** |
> | Majority Vote. (LLaMA-3 8B) | 52.54 | 74.33 | 17.06 | 12.60 | 62.41 | 52.53 |
> | IAP-mv (LLaMA-3 8B) | **66.34 (+13.80)** | **77.33 (+3.00)** | **19.25 (+2.19)** | **42.40 (+29.80)** | **68.39 (+5.98)** | **59.65 (+7.12)** |
>
> The results show that the majority vote approach performs poorly, and our IAP-mv outperforms it by a large margin, demonstrating that most prompts can lead the LLM to generate wrong answers for a given question, reaching only a few correct answers, i.e., such methods cannot recognize good/bad reasoning. Our IAP-mv can handle that with the analysis for information flow in reasoning, i.e., IAP-mv can differentiate good and bad reasoning, validating the effectiveness of our proposed strategy, rather than benefiting from multiple reasoning paths.
>
>
> **Weakness 3:**
> Some experimental conclusions are missing or somewhat unreasonable.
>
> **Response to Weakness 3:**
> We apologize for not providing an extensive explanation of the consistency and complementary experiments, we have discussed adding the following part to the 1st paragraph of Section 4.3 in our new manuscript:
>
>    > We can observe that the great performance of the instructive group outperforms our groups, which comes from the base of all instructive prompts. Furthermore, the combination of the instructive group and the other two can continue to improve the performance of both, demonstrating that complementary is critical for IAP-mv, and IAP-mv can take advantage of these complementary prompts.
>
> In the Efficacy part, we intended to express that IAP-ss and IAP-mv can be chosen with different time budgets, and we are sorry for selecting an improper figure to confuse you. In fact, the IAP-ss outperforms all the task-level optimal prompts of most LLMs and datasets, as shown in Table 1, showing that people can still employ different strategies to achieve better zero-shot prompting. We have also conducted other time experiments, and plan to replace the current figure with the new one in the updated manuscript.
>
> For some common questions, we made a unified reply in the Author Rebuttal part (see at top) which you can look up.

---

> > ### Comment · Reviewer_43EH · 2024-08-09
> > **Response to Rebuttal by Authors**
> >
> > Thank you for your responses.
> > After reading your rebuttal, I still have the following two major concerns:
> >
> > (1) IAP-ss needs to obtain the corresponding threshold on the training set. For the realistic inference scenarios, there is often no training set for parameter tuning, making the practical application scenarios of IAP-ss very limited.
> >
> > (2) The experiment setup for comparing self-consistency. The rebuttal does not provide a specific implementation explanation for self-consistency, which makes me confused. I hope the authors can give further explanations.

---

> > > ### Author Response · Authors · 2024-08-10
> > >
> > > Thanks a lot for your instant reply, and we are glad to respond to your concerns.
> > >
> > > ### **Concern 1**
> > > Your question is of practice value, IAP-ss needs to search thresholds with corresponding training sets, however, we want to briefly retrospect IAP, and provide detailed explanations to your question.
> > >
> > > IAP consists of IAP-ss and IAP-mv, IAP-mv aims to select the top-k prompts with highest saliency scores. Though k is also a hyperparameter, it is a relatively discrete number and is more easily obtained (the simple way is to observe it directly from Figure 4 in the paper). Besides, as mentioned in our previous rebuttal, under our setup of 9 prompts combination, for different datasets, k can be set to 3 to achieve consistent improvements. Hence, extending IAP-mv to real scenarios is relatively straightforward.
> > >
> > > Regarding your IAP-ss question, we provide an idea of threshold transfer, i.e., using an existing threshold to other datasets. To this end, we conducted IAP-ss experiments with GSM8K threshold to verify the transferability of thresholds.
> > >
> > > | threshold (LLaMA-3 8B) | GSM8K | SVAMP | C-Judge. | T-Obj. | CSQA | MMLU |
> > > | ---- | ---- | ---- | ---- | ---- | ---- | --- |
> > > | single optimal prompt | 64.52 | 76.00 | 16.04 | 40.00 | 64.95 | 55.79 |
> > > | from own training set | 66.43 | **77.33** | 16.57 | 38.80 | **65.68** | **56.49** |
> > > | from GSM8K training set | **66.43** | 74.00 | **17.64** | **40.80** | 64.95 | 55.09 |
> > >
> > > In the above table, we assumed all other datasets are real-world scenarios (without training sets, no knowledge of task-level best prompt), and we can see results based on GSM8K threshold, approaching or even surpassing the results of other datasets at their own thresholds (this is reasonable as threshold is successive value which usually does not cover a large tuning range; also, under few-shot prompt scenarios, demonstrations from GSM8K are commonly transferred in other datasets, indicating the adaptivity for other datasets) or under best prompts though we may not know which prompt is task-level best. Therefore, we can conclude that IAP-ss is still a potential choice without training sets.
> > >
> > > Furthermore, we recommend in practical scenarios, one can choose a dataset setting more similar to specific contexts, or draw on methods such as online learning.
> > >
> > > ### **Concern 2**
> > > We are sorry for not explaining more details of zero-shot Majority Vote (short for Zero-shot-mv) experiments earlier. We would like to first explain the details of the Zero-shot-mv experiments in former rebuttals, then clarify the self-consistency (SC) [4] you mentioned. At last, we extend supplementary experiments and discuss further.
> > >
> > > As introduced in former rebuttals, Zero-shot-mv performs inference on 9 prompts individually and then uses majority voting based on the 9 results, which is a common ensemble method. You mentioned SC uses majority voting, but it expands decoding reasoning paths by modifying greedy sampling, and we refer to it as SC-mv. Their paper also stated: "Self-consistency is completely compatible with other ensemble strategies", shown in Table 7 in the paper. In addition, the papers you cited before(for example, [1][2]) further expand the prompt diversity, which is also compatible with Zero-shot-mv.
> > >
> > > Now, back to Zero-shot-mv, we conducted additional experiments based on the suggestions of **Reviewer WuYd**, and we would like to make a deeper explanation for you. We conducted Zero-shot-mv with the other 7 zero-shot prompts (#1-7) first, and further selected top-3 prompts with the highest accuracy (fixed 3 prompts) for corresponding LLMs and datasets.
> > >
> > > | Method (LLaMA-3 8B) | GSM8K | SVAMP | C-Judge. | T-Obj. | CSQA | MMLU |
> > > | ------ | ----- | ----- | -------  | ------ | ---- | ---- |
> > > | single optimal prompt | 64.52 | 76.00 | 16.04 | 40.00 | 64.95 | 56.67 |
> > > | Zero-shot-mv (all prompts) | 52.54 | 74.33 | *17.06* | 12.60 | 62.41 | 52.53 |
> > > | Zero-shot-mv (#1-7) | 57.82 | *77.00* | *18.13* | 20.80 | *65.03* | 41.23 |
> > > | Zero-shot-mv (fixed 3 prompts) | *65.10* | *76.67* | *18.72* | 33.60 | *67.65* | *56.84* |
> > > | IAP-mv | **66.34** | **77.33** | **19.25** | **42.40** | **68.39** | **59.65** |
> > >
> > > The **bolded numbers** are best results, *italic* are results outperform task-level optimal prompts.
> > >
> > > Taking LLaMA-3 8B as an example, the results of Zero-shot-mv (#1-7) improved a lot compared to all prompts, indicating some prompts were harmful to consistency. Also, Zero-shot-mv (fixed 3 prompts) surpassed best task-level single prompts on most datasets, at least comparable, showing with better prompts combination, the Zero-shot-mv can improve further. However, IAP-mv still outperforms all, demonstrating IAP-mv can select instance-adaptive good prompts by analyzing the information flow, which is more adaptive and effective than fixed Zero-shot-mv. We also depicted a schematic case in the discussions with **Reviewer WuYd** to illustrate that, and you can refer to that.
> > >
> > > [4] Self-Consistency Improves Chain-of-Thought Reasoning.

---

> ### Comment · Area_Chair_KsdM · 2024-08-07
>
> Thank you for your review,
>
> The authors responded to your initial review. Please be sure to read it and reply indicating the extent to which the authors have addressed your initial questions and concerns.
>
> Best,
>
> AC

---

> ### Comment · Reviewer_43EH · 2024-08-10
>
> Thanks for your detailed explanation. My major concern2 has been solved, and I have already  corresponding adjusted my score.

---

> > ### Author Response · Authors · 2024-08-10
> >
> > Thanks a lot for your instant reply, we are glad to hear that our explanations addressed your concerns, and we truly appreciate your input.

---

### Official Review · Reviewer_WuYd · 2024-07-04

**Soundness:** 3
**Presentation:** 2
**Contribution:** 3
**Rating:** 6
**Confidence:** 4

**Summary:**

The paper introduces an instance-adaptive prompting algorithm for zero-shot Chain-of-Thought (CoT) reasoning in large language models (LLMs). Traditional task-level prompts are insufficient for all instances, so the authors propose a strategy that differentiates good and bad prompts based on information flow from the question to the prompt and rationale. Using neuron saliency score analysis, the study reveals that successful reasoning requires prompts to aggregate semantic information from the question. The proposed instance-adaptive prompting strategy (IAP) demonstrates consistent improvements across multiple reasoning tasks with LLaMA-2, LLaMA-3, and Qwen models.

**Strengths:**

- **Comprehensive Analysis**: Uses neuron saliency scores to understand information flow during reasoning.
- **Innovative Approach**: Tailors prompts to individual instances, improving upon uniform task-level prompts.

**Weaknesses:**

- **Limited Scope of Neuron Saliency Score Analysis**: The neuron saliency score analysis in Section 2 is conducted on only one LLM (not state in paper, personal guess) and one dataset, GSM8K. More evidence and diverse datasets are needed to support this analysis comprehensively. Additionally, Section 2.3, "Head Analysis," lacks a comparative study between good and bad reasoning instances.
- **Insufficient Experimental Details**: The experimental details provided are not sufficiently clear. Based on lines 263-264, my understanding is that IAP uses 9 different zero-shot CoT prompts to compute the S score. For IAP-ss, the process stops and uses the current result upon encountering the first prompt that meets the threshold. For IAP-mv, results from all 9 prompts are saved, and the top K are used for voting. This raises several questions: 1. How does the performance of IAP-mv compare to directly using a majority vote across the 9 prompt results? 2. What is the distribution of results for each method? For instance, which prompts tend to have higher S scores? This analysis could reveal conclusions such as "Let’s think step by step" being more suitable for math questions, while "Don’t think. Just feel." may be better for MMLU, aligning more closely with the instance-wise topic.
- **Figure 1 Inconsistency**: Figure 1 does not align with the rest of the paper. The figure and its description suggest addressing an overly complex problem, which is not the case in practice.
- **More Model Variants**: There is a need for experiments on larger model sizes to validate the findings.
- **Terminology Inconsistency**: The terminology used is inconsistent. In Figure 3, the terms "Good reasoning" and "Bad reasoning" are used, while in Figure 4, the terms change to "Good prompt" and "Bad prompt."

**Questions:**

N/A

---

> ### Author Rebuttal · Authors · 2024-08-07
>
> Thanks a lot for your valuable reviews, and we appreciate the time and effort you have taken. Regarding the weaknesses and questions, we would like to elaborate and address your concerns on this work.
>
> ### **Response to Weaknesses**
>
> **Weakness 1:**
> Limited Scope of Neuron Saliency Score Analysis: The neuron saliency score analysis in Section 2 is conducted on only one LLM (not stated in the paper, personal guess) and one dataset, GSM8K. More evidence and diverse datasets are needed to support this analysis comprehensively. Additionally, Section 2.3, "Head Analysis," lacks a comparative study between good and bad reasoning instances.
>
> **Response to Weakness 1:**
> In our investigation, we analyzed the information flow with the same method in different LLMs on various datasets and found that the phenomena of saliency scores for all LLMs on most datasets are quite similar, so we put the analysis process of Qwen-14B on GSM8K to maintain consistency in the narrative subject and present our analysis conclusion. Similarly, the head analysis results for good and bad reasoning are consistent, we are sorry for not elaborating on that in this subsection. We apologize for not putting more figures in the Appendix to make our analysis more complete and well-supported, and we have updated our manuscript. We also put a few samples of the analysis process in the pdf file attached in the Author Rebuttal (see at top).
>
>
> **Weakness 2:**
> Insufficient Experimental Details
>
> **Response to Weakness 2:**
> Your understanding of IAP is correct, and your suggestion of supplementing answers to the majority vote experiment would make our work more complete. The results comparison between our IAP-mv and the direct answers majority vote is as follows:
>
> | Method | GSM8K | SVAMP | C-Judge. | T-Obj. | CSQA | MMLU |
> | ------ | ----- | ----- | -------  | ------ | ---- | ---- |
> | Majority Vote. (Qwen 14B) | 28.22 | 51.33 | 26.10 | 1.44 | 46.52 | 52.46 |
> | IAP-mv (Qwen 14B) | **62.81 (+34.59)** | **73.33 (+22.00)** | **29.95 (+3.85)** | **25.60 (+24.16)** | **65.68 (+19.16)** | **78.95 (+26.49)** |
> | Majority Vote. (LLaMA-3 8B) | 52.54 | 74.33 | 17.06 | 12.60 | 62.41 | 52.53 |
> | IAP-mv (LLaMA-3 8B) | **66.34 (+13.80)** | **77.33 (+3.00)** | **19.25 (+2.19)** | **42.40 (+29.80)** | **68.39 (+5.98)** | **59.65 (+7.12)** |
>
> The results show that the majority vote approach performs poorly, and our IAP-mv outperforms it by a large margin, demonstrating that most prompts can lead the LLM to generate wrong answers for a given question, reaching only a few correct answers, i.e., such methods cannot recognize good/bad reasoning. Our IAP-mv can handle that with the analysis for information flow in reasoning, i.e., IAP-mv can differentiate good and bad reasoning, validating the effectiveness of our proposed strategy.
>
> The most adopted prompts in IAP-mv of distinct LLMs on different datasets are the former task-level optimal (among all prompt candidates) since they are suitable for most instances in corresponding datasets, respectively. We display the top-3 prompts for different LLMs on different datasets in the table below:
>
> | Model | GSM8K | SVAMP | C-Judge. | T-Obj. | CSQA | MMLU |
> | ------ | ----- | ----- | -------  | ------ | ---- | ---- |
> | LLaMA-3 8B | #1, 6, 7 | #1, 2, 6 | #2, 4, 8 | #1, 2, 4 | #2, 4, 6 | #1, 2, 6 |
> | Qwen 14B | #1, 3, 6 | #3, 4, 6 | #2, 3, 5 | #2, 3, 5 | #1, 6, 9 | #2, 4, 5 |
>
> We can observe that the most adopted prompt candidates are the former task-level optimal ones, which is consistent with our analysis in the paper.
>
>
> **Weakness 3:**
> Figure 1 Inconsistency: Figure 1 does not align with the rest of the paper.
>
> **Response to Weakness 3:**
> Figure 1 offered a representative case to illustrate that: the worst task-level prompt can beat the optimal prompt for some instances, which is counterintuitive as we discussed in the 2nd paragraph in the Introduction section, such a case encouraged us to detect the inner mechanism for zero-shot CoT, further trigged us to propose instance-level prompting strategy.
>
>
> **Weakness 4:**
> More Model Variants.
>
> **Response to Weakness 4:**
> In this paper, we conducted experiments on 8B, 13B, and 14B LLMs with different architectures (LLaMA-2, LLaMA-3, and Qwen), and the results validated our initiative observation and analysis. The current evaluation across model sizes has provided a broad view of how different scales of models may perform under various prompt candidates. However, your suggestion is also valuable, and we conducted the experiments with LLaMA-3 70B using the same 9 prompts as the main experiments and IAP-mv, the table below shows the results:
>
> | Prompt | GSM8K | SVAMP | C-Judge. | T-Obj. | CSQA | MMLU |
> | ------ | ----- | ----- | -------  | ------ | ---- | ---- |
> | #1 | 87.79 | 82.33 | 38.50 | 12.40 | 67.73 | 37.02 |
> | #2 | 89.16 | 86.33 | 54.55 | 30.00 | 56.10 | 50.18 |
> | #3 | 81.73 | 83.33 | 49.73 | 23.20 | 55.69 | 44.56 |
> | #4 | 82.64 | 84.33 | 42.25 | 60.40 | 41.36 | 52.11 |
> | #5 | 82.71 | 84.00 | 36.36 | 6.80 | 61.75 | 52.63 |
> | #6 | 87.79 | 82.33 | 44.39 | 16.00 | 67.73 | 35.79 |
> | #7 | 81.43 | 85.67 | 47.59 | 24.00 | 29.98 | 14.56 |
> | #8 | 53.53 | 75.67 | 55.61 | 18.40 | 29.24 | 22.56 |
> | #9 | 51.71 | 58.33 | 44.92 | 20.40 | 36.94 | 43.33 |
> | IAP-mv | **89.84** | **87.33** | **56.20** | **62.00** | **69.04** | **54.39** |
>
> In this table, IAP-mv demonstrates its effectiveness on LLaMA-3 70B, consistent with the results on LLaMA-3 8B and Qwen 14B, broadening the model scale impact of the IAP-mv.
>
>
> **Weakness 5:**
> In Figure 3, the terms "Good reasoning" and "Bad reasoning" are used, while in Figure 4, the terms change to "Good prompt" and "Bad prompt."
>
> **Response to Weakness 5:**
> We are sorry for not using the same statement in different figures, and we have rectified such issues to keep the claims consistent in the updated manuscript.
>
> For some common questions, we made a unified reply in the Author Rebuttal part (see at top) which you can look up.

---

> > ### Comment · Reviewer_WuYd · 2024-08-08
> >
> > Thank you for the authors' response. I have reviewed your feedback and noted that most of our concerns have been addressed.
> > However, I observed an issue with your supplementary experiment on IAP-mv VS majority voting, which shows unusual performance. According to Table 1 in the paper and the new table, nearly every score for Majority Vote is around the lower bound of the baseline scores in Table 1 with 9 prompt candidates. In my personal experience, the Majority Vote should not exhibit this behavior. Therefore, I am wondering if there might be a mistake in your experimental setup or if prompts #8 and #9 negatively influence the Majority Vote method. Could you please check the Majority Vote experiment again and provide updated scores excluding prompts #8 and #9?

---

> > > ### Author Response · Authors · 2024-08-08
> > >
> > > Thanks a lot for your instant reply, and we are glad to hear that most of your concerns were addressed.
> > >
> > > You raised a great question, the majority vote is quite an important tool and we have checked the settings again for the supplementary experiments. Your judgment is right, zero-shot prompts-based majority vote (short for Zero-shot-mv) has some unusual results, raising by involving extra bad prompts (#8 and #9). According to your advice, we conducted experiments with the other 7 zero-shot prompts (#1-7), the results were enhanced, and we further selected top-3 prompts with the highest accuracy (top-3 prompts) for corresponding LLMs and datasets. With a better prompt combination, the performance of the Zero-shot-mv could be promoted. Taking LLaMA-3 8B as an example, Zero-shot-mv (with top-3 prompts) surpassed the best task-level single zero-shot prompts on most datasets, at least comparable. However, IAP-mv still outperforms all, demonstrating that IAP-mv can select instance-adaptive good prompts by analyzing the information flow, which is more adaptive and effective than the fixed Zero-shot-mv. Next, we will provide a detailed explanation of the experiments.
> > >
> > > | Method (LLaMA-3 8B) | GSM8K | SVAMP | C-Judge. | T-Obj. | CSQA | MMLU |
> > > | ------ | ----- | ----- | -------  | ------ | ---- | ---- |
> > > | single optimal prompt | 64.52 | 76.00 | 16.04 | 40.00 | 64.95 | 56.67 |
> > > | Zero-shot-mv (all prompts) | 52.54 | 74.33 | *17.06* | 12.60 | 62.41 | 52.53 |
> > > | Zero-shot-mv (#1-7) | 57.82 | *77.00* | *18.13* | 20.80 | *65.03* | 41.23 |
> > > | Zero-shot-mv (fixed 3 prompts) | *65.10* | *76.67* | *18.72* | 33.60 | *67.65* | *56.84* |
> > > | IAP-mv | **66.34** | **77.33** | **19.25** | **42.40** | **68.39** | **59.65** |
> > >
> > > | Method (Qwen 14B) | GSM8K | SVAMP | C-Judge. | T-Obj. | CSQA | MMLU |
> > > | ------ | ----- | ----- | -------  | ------ | ---- | ---- |
> > > | single optimal prompt | 60.50 | 72.00 | 28.34 | 23.20 | 63.23 | 76.84 |
> > > | Zero-shot-mv (all prompts) | 28.22 | 51.33 | 26.10 | 1.44 | 46.52 | 52.46 |
> > > | Zero-shot-mv (#1-7) | 57.98 | 55.33 | 27.50 | 8.80 | 57.70 | 63.86 |
> > > | Zero-shot-mv (fixed 3 prompts) | *61.91* | 71.66 | *28.34* | 20.80 | **66.01** | 74.38 |
> > > | IAP-mv | **62.81** | **73.33** | **29.95** | **25.60** | *65.68* | **78.95** |
> > >
> > > The **bolded numbers** are the best results, and the *italic* are the results that outperform the task-level optimal prompt.
> > >
> > >
> > > ### **Zero-shot-mv without misleading prompts**
> > > This table demonstrated that without misleading prompt candidates (#8 and #9), the performance of Zero-shot-mv can be improved compared to the all prompts-based setting, indicating that some prompts are harmful to the consistency, which is in line with your assumption. We present a schematic case to exemplify why the Zero-shot-mv(#1-7) performs better than the all prompts-based setting (supposed there are only two answers, right or wrong):
> > >
> > > | Method | #1 | #2 | #3 | #4 | #5 | #6 | #7 | #8 | #9 |
> > > | --- | --- | --- | --- | --- | --- | --- | --- | --- | --- |
> > > | Zero-shot-mv(all prompts) | &#10004; | &#10004; | &#10007; | &#10004; | &#10007; | &#10007; | &#10004; | &#10007; | &#10007; |
> > > | Zero-shot-mv(#1-7) | &#10004; | &#10004; | &#10007; | &#10004; | &#10007; | &#10007; | &#10004; |  |  |
> > >
> > > We can see that the Zero-shot-mv(all prompts) gets the wrong answer because the number of wrong answers is more. For the Zero-shot-mv(#1-7), it can get the correct result with the answers ensemble after we eliminate 2 misleading prompts.
> > >
> > > ### **Zero-shot-mv with fixed 3 prompts**
> > > We further testified the top-3 prompts with the highest accuracy of corresponding LLMs and datasets (fixed 3 prompts), and we found that with better prompt candidates, the Zero-shot-mv can improve further. Even though, the IAP-mv still outperforms Zero-shot-mv(fixed 3 prompts), the IAP-mv can take advantage of both consistency and complementary among those prompts at the instance level. Here, we present another schematic case to exemplify why the IAP-mv can outperform Zero-shot-mv(fixed 3 prompts, #1-3 in the table below, suppose there are only two answers, right or wrong):
> > >
> > > | Method | #1 | #2 | #3 | #4 | #5 | #6 | #7 | #8 | #9 |
> > > | --- | --- | --- | --- | --- | --- | --- | --- | --- | --- |
> > > | Zero-shot-mv(all prompts) | &#10004; | &#10007; | &#10007; | &#10007; | &#10007; | &#10007; | &#10004; | &#10004; | &#10007; |
> > > | Zero-shot-mv(fixed 3 prompts) | &#10004; | &#10007; | &#10007; |  |  |  |  |  |  |
> > > | IAP-mv | &#10004; | &#10007; |  |  |  |  |  | &#10004; |  |
> > >
> > > In this case, the IAP-mv can get the correct answer with more-grained recognition for suitable prompts dynamically in the instance level, thereby outperforming the fixed 3 prompts.
> > >
> > > Furthermore, the Zero-shot-mv depends on picking fixed prompts, requiring prior knowledge and inflexibility, thus such methods are not easy to obtain in a real-world scenario. In contrast, IAP-mv is more generalizable and can be applied to any new task adaptively without handcraft selection.

---

> > > > ### Comment · Reviewer_WuYd · 2024-08-08
> > > >
> > > > Thank you for the authors' response.
> > > > I feel that your response has addressed my concerns and provided me with a deeper understanding of your paper. I thoroughly enjoyed our discussion stage; it felt like a very productive academic exchange! I will be raising my score. I hope the authors can expedite the open-sourcing of the code for this work and provide an easy-to-use repository. I am very much looking forward to it!

---

> > > > > ### Author Response · Authors · 2024-08-09
> > > > >
> > > > > Thank you very much for your valuable comments, we also enjoyed the process of responding to your questions and really gained a lot from that! We will release the algorithm and experimental code once this paper is accepted.

---

> ### Comment · Area_Chair_KsdM · 2024-08-07
>
> Thank you for your review,
>
> The authors responded to your initial review. Please be sure to read it and reply indicating the extent to which the authors have addressed your initial questions and concerns.
>
> Best,
>
> AC

---

### Official Review · Reviewer_UGUn · 2024-07-09

**Soundness:** 4
**Presentation:** 4
**Contribution:** 3
**Rating:** 8
**Confidence:** 4

**Summary:**

This paper analyzed the mechanism of the large language models (LLMs) zero-shot Chain-of-Thought (CoT) reasoning, in which the authors found a pattern to discriminate a good reasoning path and a bad one with the saliency scores. Based on the findings, this paper proposed a set of instance-adaptive prompting approaches for zero-shot CoT reasoning. Experimental results on various tasks with distinct LLMs demonstrated the effectiveness of the proposed methods, validating the correctness of the findings.

**Strengths:**

1. This paper is well-written and easy to follow.
2. This paper emphasized the instance-level CoT prompting rather than the task-level, providing a novel and fine-grained research object for LLM reasoning, which had not been explored in earlier works.
3. This work employed an interesting information flow analysis to delve into the mechanism of zero-shot CoT during the LLM inference, encouraging relevant CoT research.
4. The authors presented two simple yet effective instance-adaptive zero-shot CoT prompting approaches based on elaborative analysis, and empirical results verified their observations.

**Weaknesses:**

1. Experiments only covered 7B and 13/14B models, involving larger model in the experiments would be more marvelous.
2. In Section 4.3, the authors didn’t interpret the results of consistency and complementary results in detail.

**Questions:**

1. Why did the authors choose the saliency score instead of the widely used attribution score as the information flow analysis method?
2. What did some results with underlines in Table 1 mean?
3. Should the colon after the “remains” in the line 44 be removed?
4. Should the “and” in the “, and Tracking Shuffled Objects …” in line 258 be removed?
5. In Table 3 in the Appendix, why didn’t the authors highlight the best results and explain the results more?

All my concerns have been addressed by the authors' rebuttal.

**Limitations:**

The authors have adequately clarified the limitations of this work.

---

> ### Author Rebuttal · Authors · 2024-08-07
>
> Thanks a lot for the time and effort you invested in providing the detailed reviews. Regarding the current weaknesses and questions you pointed out, we are glad to give our responses.
>
> ### **Response to Weaknesses**
> **Weakness 1:**
> Experiments only covered 7B and 13/14B models, involving larger models in the experiments would be more marvelous.
>
> **Response to Weakness 1:**
> In this work, we conducted experiments on 8B, 13B, and 14B LLMs with different architectures (LLaMA-2, LLaMA-3, and Qwen), and empirical results validated our initiative observation and assumption. The current evaluation across model sizes has provided a broad view of how different scales of models may perform under various prompt candidates.
>
> We conducted the experiments with LLaMA-3 70B using the same 9 prompts as the main experiments and IAP-mv, the table below shows the results:
>
> | Prompt | GSM8K | SVAMP | C-Judge. | T-Obj. | CSQA | MMLU |
> | ------ | ----- | ----- | -------  | ------ | ---- | ---- |
> | #1 | 87.79 | 82.33 | 38.50 | 12.40 | 67.73 | 37.02 |
> | #2 | 89.16 | 86.33 | 54.55 | 30.00 | 56.10 | 50.18 |
> | #3 | 81.73 | 83.33 | 49.73 | 23.20 | 55.69 | 44.56 |
> | #4 | 82.64 | 84.33 | 42.25 | 60.40 | 41.36 | 52.11 |
> | #5 | 82.71 | 84.00 | 36.36 | 6.80 | 61.75 | 52.63 |
> | #6 | 87.79 | 82.33 | 44.39 | 16.00 | 67.73 | 35.79 |
> | #7 | 81.43 | 85.67 | 47.59 | 24.00 | 29.98 | 14.56 |
> | #8 | 53.53 | 75.67 | 55.61 | 18.40 | 29.24 | 22.56 |
> | #9 | 51.71 | 58.33 | 44.92 | 20.40 | 36.94 | 43.33 |
> | IAP-mv | **89.84** | **87.33** | **56.20** | **62.00** | **69.04** | **54.39** |
>
> The empirical results of the LLaMA-3 70B with IAP-mv are better than any task-level hard prompt, which demonstrates the effectiveness of the IAP-mv under broader model scales.
>
>
> **Weakness 2:**
> In Section 4.3, the authors didn’t interpret the results of consistency and complementary results in detail.
>
> **Response to Weakness 2:**
> In the ablation studies, we conducted the consistency and complementary experiment to detect which type of prompt combination contributed to the performance. Table 2 shows the results, and referring to the main results in Table 1, we can observe that each pair of combinations can improve the performance, and instructive and irrelevant combinations achieve better outcomes than others, which comes from the base performance of instructive prompts.
>
>
> ### **Response to Questions**
> **Question 1:**
> Why did the authors choose the saliency score instead of the widely used attribution score as the information flow analysis method?
>
> **Response to Question 1:**
> There are two reasons that we didn't employ the attribution score: (1) the attribution score is designed for knowledge-oriented tasks, which have been discussed in the related work section; (2) the saliency score is more suitable for the in-context learning scenario, and may not perform well on the reasoning tasks. These two reasons had been demonstrated by [1][2][3][4].
> [1] Dai, D., L. Dong, Y. Hao, et al. Knowledge neurons in pretrained transformers.
> [2] Hao, Y., L. Dong, F. Wei, et al. Self-attention attribution: Interpreting information interactions inside transformer.
> [3] Wang, L., L. Li, D. Dai, et al. Label words are anchors: An information flow perspective for understanding in-context learning.
> [4] Li, J., P. Cao, C. Wang, et al. Focus on your question! interpreting and mitigating toxic cot problems in commonsense reasoning.
>
> **Question 2:**
> What did some results with underlines in Table 1 mean?
>
> **Response to Question 2:**
> In Table 1, we underlined some numbers which means these results are task-level optimal prompts among all candidates, we are sorry for not elaborating on this, and we have explained in the new version.
>
> **Question 3:**
> Should the colon after the “remains” in line 44 be removed?
>
> **Response to Question 3:**
> Thank you for your detailed review, we have removed the colon after the “remains” in line 44 in the updated manuscript.
>
> **Question 4:**
> Should the “and” in the “, and Tracking Shuffled Objects …” in line 258 be removed?
>
> **Response to Question 4:**
> We have removed the “and” in the “, and Tracking Shuffled Objects …”, and we have checked all the grammar issues in the new manuscript.
>
> **Question 5:**
> In Table 3 in the Appendix, why didn’t the authors highlight the best results and explain the results more?
>
> **Response to Question 5:**
> Since Table 3 in the Appendix is the supplement of Table 1, and the IAP shows similar performance on LLaMA-2 13B with LLaMA-3 8B and Qwen 14B, thus we didn’t repeat similar explanations. Also, we have highlighted the best results as we did in Table 1 in the updated manuscript.
>
> For some common questions, we made a unified reply in the Author Rebuttal part (see at top) which you can look up.

---

> > ### Comment · Reviewer_UGUn · 2024-08-10
> > **Response to the authors' rebuttal**
> >
> > Thank you for the authors' responses. I have read the rebuttal and all my concerns have been addressed. This paper analyzed the inner machanism of zero-shot CoT and proposed a novel instance-adaptive prompting strategy. I also reviewed the comments from the other reviewers, and believe this is quite a solid work. I will raise the score from 7 to 8.

---

> > > ### Author Response · Authors · 2024-08-13
> > >
> > > Thanks a lot for your valuable reply, we appreciate the time and effort you invested, and we are so glad to know that your concerns have been addressed.

---

> ### Comment · Area_Chair_KsdM · 2024-08-07
>
> Thank you for your review,
>
> The authors responded to your initial review. Please be sure to read it and reply indicating the extent to which the authors have addressed your initial questions and concerns.
>
> Best,
>
> AC

---

### Official Review · Reviewer_dc7a · 2024-07-11

**Soundness:** 3
**Presentation:** 2
**Contribution:** 2
**Rating:** 7
**Confidence:** 4

**Summary:**

The authors argue that a single, task-level prompt is insufficient for addressing the diverse needs of different instances within a dataset. To overcome this limitation, they propose an instance-adaptive prompting (IAP) algorithm that differentiates between effective and ineffective prompts for individual instances. The authors also provide a detailed examination of the information flow at different layers and heads of the LLMs, offering insights into the internal mechanisms that contribute to reasoning quality. The proposed IAP strategy is shown to be effective across multiple models and tasks, highlighting its potential for advancing zero-shot reasoning in LLMs.

**Strengths:**

1. The IAP algorithm is a creative solution that addresses the limitations of previous prompt strategy. The originality lies in the adaptive differentiation of prompts for individual instances and the use of information flow analysis to understand and enhance reasoning mechanisms within LLMs.

2. The quality of the experimental design is high, with comprehensive testing across different models and reasoning tasks.

**Weaknesses:**

1. As the number of available prompts grows, the strategy for selecting the best prompt could become increasingly complex.

2. The titles of some figures are verbose (e.g. Figure 2). Keeping the title concise and giving a detailed explanation of the figures in the main text will make the paper better.

**Questions:**

1. How were the saliency score thresholds determined, how do these thresholds affect the results?

2. In some paragraphs, full stops are missing, for example, at the end of the "Experiments" section and at the end of the first paragraph in the "Preliminary analysis" section.

**Limitations:**

it appears that the authors have made an effort to address the limitations of their work.

---

> ### Author Rebuttal · Authors · 2024-08-07
>
> We deeply appreciate the time and effort you invested in reviewing our paper, and we are glad to answer your questions.
>
> ### **Responses to Weaknesses**
>
> **Weakness 1:**
> As the number of available prompts grows, the strategy for selecting the best prompt could become increasingly complex.
>
> **Response to Weakness 1:**
> For IAP-ss, we obtained threshold values with regard to distinct LLMs on different training sets, we computed the overall synthesized scores (defined in eq (4) in Section 3) to divide up the good and bad reasoning paths and adopted the thresholds whose classify reasoning well. Such as, the threshold of LLaMA-3 8B on GSM8K is 5.5e-6, and the identification of the thresholds of different LLMs on different datasets is the same and it is simple and doesn't not need much time. In practice, we considered reasoning with a value higher than the threshold as good, otherwise bad. We have tried different thresholds, and the best performance is shown in the following table.
>
> | threshold | accuracy |
> | --- | --- |
> | 7.0e-6 | 59.82 |
> | 6.0e-6 | 62.77 |
> | 5.0e-6 | 64.67 |
> | 4.0e-6 | 62.40 |
> | 5.5e-6 | **65.36** |
>
> We can see that an improper threshold can affect the performance of the IAP-ss, no matter higher or lower. It comes from that higher thresholds tend to recognize some good reasoning instances as bad ones, and lower thresholds may overlook some bad reasoning.
>
> As we discussed in the last paragraph in Section 3 and Section 4, IAP-mv can achieve better performance with more computing, while IAP-ss can be a choice under the resource-limited scenario, and it would be faster if the order of prompt candidates were arranged properly. We conducted the time experiments with LLaMA-3 8B on SVAMP under different orders of prompt candidates, and the results are shown in the table below (The numbers in the order column represent the prompt candidates as we introduced in Section 4.1 Zero-shot CoT prompts paragraph.):
>
> | order | accuracy | time |
> | --- | --- | --- |
> | #9  | 39.67 | 2860s |
> | #6  | 76.00 | **2657s** |
> | #9, 8, 5, 4, 3  | 63.66 | 3870s |
> | #6, 1, 2, 7, 3 | 76.66 | 5216s |
> | #1, 2, 3, 4, 5, 6, 7, 8, 9 | **77.33** | 4751s |
>
> Where #9 is the worst task-level prompt, and #6 is the best task-level prompt, achieving the highest accuracy among all the prompt candidates while consuming the least time. The prompt order of the 3rd row is accuracy-decreased on SVAMP, the 4th row is accuracy-increased, and the last row is our default setting, which obtains the best performance. This table shows that IAP-ss can cost less time with fewer prompt candidates but may obtain limited results, however, even fewer improper candidates could take a lot of computing time. Therefore, we think that the time cost of IAP-ss is not a major issue if prompt candidates are in an appropriate order.
>
>
> **Weakness 2:**
> The titles of some figures are verbose (e.g. Figure 2).
>
> **Response to Weakness 2:**
> Thanks for your meticulous reviews and suggestions, we have moved that verbose explanation from the figures to the main body in the new version. Now, the explanation of Figure 2 and Figure 4 are as follows:
>
>    > (Figure 2) The visualization comparison of the saliency matrices between good and bad reasoning instances with two prompts, the darker the color of the pixel point in the image represents a larger saliency score. (a) and (b) are good and bad reasoning instances under "Let's think step by step.", and so as (c) and (d) under "Don't think. Just feel.", respectively. The red, blue, and green boxes in each subfigure depict the question-to-prompt, question-to-rationale, and prompt-to-rationale information flow, respectively.
>
>    > (Figure 4) Saliency scores of question-to-prompt, question-to-rationale, and prompt-to-rationale across layers. The yellow lines represent prompts that effectively guide the LLMs to generate the correct answer, indicating good prompts. Conversely, the blue lines denote ineffective prompts.
>
> And we have moved the previous definitions of the saliency matrices to the main body in the new version to make the presentation more clear.
>
> ### **Response to Questions**
>
> **Question 1:**
> How were the saliency score thresholds determined, and how do these thresholds affect the results?
>
> **Response to Question 1:**
> For the IAP-ss, we obtained threshold values with regard to distinct LLMs on different training sets, we computed the overall synthesized scores (defined in eq (4) in Section 3) to divide up the good and bad reasoning paths and adopted the thresholds whose classify reasoning well. In practice, we considered reasoning with a value higher than the threshold is good, otherwise bad. From this, it is evident that an improper threshold may lead to inaccurate judgment for good and bad reasoning, thereby hurting the overall performance. The current threshold of LLaMA-3 8B on GSM8K we adopted is 5.5e-6, we also tried other thresholds while keeping other settings, and the results are shown in the table below.
>
> Table caption LLaMA 3 on GSM8k, different thresholds from repeating the experiment on other LLMs and datasets. (validation selection, fast)
>
> | threshold | accuracy |
> | --- | --- |
> | 7.0e-6 | 59.82 |
> | 6.0e-6 | 62.77 |
> | 5.0e-6 | 64.67 |
> | 4.0e-6 | 62.40 |
> | 5.5e-6 | **65.36** |
>
> We can see that an improper threshold can affect the performance of the IAP-ss, whether it is higher or lower. It comes from that higher thresholds tend to recognize some good reasoning instances as bad ones, and lower thresholds may overlook some bad reasoning. Therefore, a proper threshold benefits the IAP-ss a lot, and vice versa.
>
>
> **Question 2:**
> In some paragraphs, full stops are missing.
>
> **Response to Question 2:**
> We appreciate your careful review of our paper, and we have revised all full stops missing issues in the new version and checked the entire manuscript to avoid any similar issues.
>
> For some common questions, we made a unified reply in the Author Rebuttal part (see at top) which you can look up.

---

> > ### Comment · Reviewer_dc7a · 2024-08-12
> > **Response to the authors**
> >
> > Thanks to the authors for resolving my concern. I will change me score from 6 to 7.

---

> > > ### Author Response · Authors · 2024-08-12
> > >
> > > Thank you for your valuable reply, we appreciate your effort and time, and we are so glad to hear that your concerns have been addressed.

---

> ### Comment · Area_Chair_KsdM · 2024-08-07
>
> Thank you for your review,
>
> The authors responded to your initial review. Please be sure to read it and reply indicating the extent to which the authors have addressed your initial questions and concerns.
>
> Best,
>
> AC

---

### Author Rebuttal · Authors · 2024-08-07

We want to thank each reviewer for your thoughtful reviews and constructive feedback on our manuscript. We appreciate the time you invested in evaluating our submission, and we are grateful for your detailed suggestions and recommendations. We have carefully considered each of your comments and have made corresponding revisions to our manuscript in response. We noticed that some reviewers raised similar questions, so we consolidated the responses first.

### **Experiment Details**
For IAP-ss, we obtained threshold values with regard to distinct LLMs on different training sets, we computed the overall synthesized scores (defined in eq (4) in Section 3) to divide up the good and bad reasoning paths and adopted the thresholds whose classify reasoning well. Such as, the threshold of LLaMA-3 8B on GSM8K is 5.5e-6, and the identification of the thresholds of different LLMs on different datasets is the same and it is simple and doesn't not need much time. In practice, we considered reasoning with a value higher than the threshold as good, otherwise bad. We have tried different thresholds, and the best performance is shown in the following table.

| threshold | accuracy |
| --- | --- |
| 7.0e-6 | 59.82 |
| 6.0e-6 | 62.77 |
| 5.0e-6 | 64.67 |
| 4.0e-6 | 62.40 |
| 5.5e-6 | **65.36** |

We can see that an improper threshold can affect the performance of the IAP-ss, no matter higher or lower. It comes from that higher thresholds tend to recognize some good reasoning instances as bad ones, and lower thresholds may overlook some bad reasoning.

As for IAP-mv, there is no need to compute thresholds for each LLM and dataset, we need to compute the overall synthesized saliency scores for all prompt candidates and recognize the good and bad reasoning with top-3 highest scores. **The IAP-mv is different from the direct answers majority vote strategy.** Specifically, the IAP-mv is to find the top-k (k=3) highest scores among all prompt candidates based on the analysis of information flow, in which the IAP-mv can recognize the correct answer from good and bad reasoning. We appreciate the reviewers' suggestion of supplementing the answers majority vote experiment, we did a comparison between our IAP-mv and the direct answers majority vote as follows:

| Method | GSM8K | SVAMP | C-Judge. | T-Obj. | CSQA | MMLU |
| ------ | ----- | ----- | -------  | ------ | ---- | ---- |
| Majority Vote. (Qwen 14B) | 28.22 | 51.33 | 26.10 | 1.44 | 46.52 | 52.46 |
| IAP-mv (Qwen 14B) | **62.81 (+34.59)** | **73.33 (+22.00)** | **29.95 (+3.85)** | **25.60 (+24.16)** | **65.68 (+19.16)** | **78.95 (+26.49)** |
| Majority Vote. (LLaMA-3 8B) | 52.54 | 74.33 | 17.06 | 12.60 | 62.41 | 52.53 |
| IAP-mv (LLaMA-3 8B) | **66.34 (+13.80)** | **77.33 (+3.00)** | **19.25 (+2.19)** | **42.40 (+29.80)** | **68.39 (+5.98)** | **59.65 (+7.12)** |

The results show that the majority vote approach performs poorly, and our IAP-mv outperforms it by a large margin, demonstrating that most prompts can lead the LLM to generate wrong answers for a given question, reaching only a few correct answers, i.e., such methods cannot recognize good/bad reasoning. Our IAP-mv can handle that with the analysis for information flow in reasoning, i.e., IAP-mv can differentiate good and bad reasoning, validating the effectiveness of our proposed strategy.


### **Explanation for Ablation Study**
We apologize for not providing an elaborative explanation of the consistency and complementary experiments, we have discussed adding the following part to the 1st paragraph of Section 4.3 in our new manuscript:

   > We can observe that the great performance of the instructive group outperforms our groups, which comes from the base of all instructive prompts. Furthermore, the combination of the instructive group and the other two can continue to improve the performance of both, demonstrating that complementary is critical for IAP-mv, and IAP-mv can take advantage of these complementary prompts.


### **Larger LLMs**
We have conducted experiments with several LLMs of 8B, 13B, and 14B in different architectures, shown and discussed in the original manuscript. To broaden the generalizability of our method, we further implemented experiments with LLaMA-3 70B on different datasets, results are shown in the table below:

| Prompt | GSM8K | SVAMP | C-Judge. | T-Obj. | CSQA | MMLU |
| ------ | ----- | ----- | -------  | ------ | ---- | ---- |
| #1 | 87.79 | 82.33 | 38.50 | 12.40 | 67.73 | 37.02 |
| #2 | 89.16 | 86.33 | 54.55 | 30.00 | 56.10 | 50.18 |
| #3 | 81.73 | 83.33 | 49.73 | 23.20 | 55.69 | 44.56 |
| #4 | 82.64 | 84.33 | 42.25 | 60.40 | 41.36 | 52.11 |
| #5 | 82.71 | 84.00 | 36.36 | 6.80 | 61.75 | 52.63 |
| #6 | 87.79 | 82.33 | 44.39 | 16.00 | 67.73 | 35.79 |
| #7 | 81.43 | 85.67 | 47.59 | 24.00 | 29.98 | 14.56 |
| #8 | 53.53 | 75.67 | 55.61 | 18.40 | 29.24 | 22.56 |
| #9 | 51.71 | 58.33 | 44.92 | 20.40 | 36.94 | 43.33 |
| IAP-mv | **89.84** | **87.33** | **56.20** | **62.00** | **69.04** | **54.39** |

The empirical results of the LLaMA-3 70B with IAP-mv are better than any task-level hard prompt, demonstrating the IAP-mv's effectiveness under broader model scales.


### **Grammar Error**
We have checked and modified the grammar errors in the original paper, and updated a new manuscript to avoid such issues, making the presentation more clear.

---

### Decision · Program_Chairs · 2024-09-25

**Decision:**

Accept (poster)

**Comment:**

The paper introduces an instance-adaptive prompting algorithm for zero-shot Chain-of-Thought (CoT) reasoning in large language models (LLMs). This includes an analysis of LLMs CoT reasoning, which identifies a way to discriminate good from bad reasoning paths using saliency scores, and proposes a method to enhance CoT reasoning by selecting an appropriate prompt from a given set in response to the specific query. In experiments, the proposed IAP strategy is shown to be effective across multiple models and tasks.

The reviewers identified a number of strengths for the paper. It is well written with an intuitive motivation. The emphasis on instance-level CoT prompting and the use of saliency scores to analyse information flow appears to be innovative and promising. The experimental design is sound, the evaluations extensive, and results are promising.

There were some concerns about the generality of the analysis and experimentation, and whether the advantages seen were at least partly a result of the fairness of the comparison. Many of the reviewer concerns were addressed by author rebuttal and further results and explanations. It is unclear how much room there will be in the final manuscript for all these additional points though. Nonetheless, the submission is of sufficiently high quality to warrant acceptance, and to garner interest.